# Strain regulates the photovoltaic performance of thick-film perovskites

Pengju Shi[1,2,10], Jiazhe Xu[1,2,10], Ilhan Yavuz [3,10], Tianyi Huang[4], Shaun Tan [4], Ke Zhao[1,2], Xu Zhang[1,2], Yuan Tian[1,2], Sisi Wang[2], Wei Fan[1,2], Yahui Li[1,2], Donger Jin[1], Xuemeng Yu[5], Chenyue Wang[6], Xingyu Gao [6], Zhong Chen[7], Enzheng Shi [1], Xihan Chen [5], Deren Yang[1], Jingjing Xue [1,8] ✉, Yang Yang [4] ✉ & Rui Wang [2,9] ✉

Perovskite photovoltaics, typically based on a solution-processed perovskite layer with a film thickness of a few hundred nanometres, have emerged as a leading thin-film photovoltaic technology. Nevertheless, many critical issues pose challenges to its commercialization progress, including industrial compatibility, stability, scalability and reliability. A thicker perovskite film on a scale of micrometres could mitigate these issues. However, the efficiencies of thick-film perovskite cells lag behind those with nanometre film thickness. With the mechanism remaining elusive, the community has long been under the impression that the limiting factor lies in the short carrier lifetime as a result of defects. Here, by constructing a perovskite system with extraordinarily long carrier lifetime, we rule out the restrictions of carrier lifetime on the device performance. Through this, we unveil the critical role of the ignored lattice strain in thick films. Our results provide insights into the factors limiting the performance of thick-film perovskite devices.

The ever-growing field of photovoltaics has witnessed the rapid success of halide perovskites in achieving a high power conversion efficiency (PCE) over 25%[1,2]. Perovskite solar cells (PSCs), typically based on a solution-processed perovskite layer with a film thickness of a few hundred nanometers, have emerged as a leading thin-film photovoltaic technology. PSCs hold great promise to be a game-changer in the photovoltaic industry, whereas many critical issues pose challenges to its commercialization progress[3–6]. Such challenges derive from the confluence of several factors, including industrial compatibility[7,8], environmental stability[9,10], scalability[4,5] and production reliability[11]. With such consideration, a thicker perovskite film on a scale of a few micrometers could mitigate these issues. Firstly, with regards to its huge commercialization potential in tandem devices, a thick perovskite film promises industrial compatibility when stacked on the top surface of rear cells with micron-scale roughness such as commercialized textured silicon[7] and copper indium gallium selenide[8]. In contrast, a perovskite film with a few hundred nanometre thickness usually fails to achieve full surface coverage of the textured rear cell.

[1]State Key Laboratory of Silicon Materials and School of Materials Science and Engineering, Zhejiang University, Hangzhou 310027, China. [2]Research Center for Industries of the Future, School of Engineering, Westlake University and Institute of Advanced Technology, Westlake Institute for Advanced Study, Hangzhou 310024, China. [3]Department of Physics, Marmara University, Ziverbey, Istanbul 34722, Turkey. [4]Department of Materials Science and Engineering and California NanoSystems Institute, University of California, Los Angeles, CA 90095, USA. [5]Department of Mechanical and Energy Engineering, Southern University of Science and Technology, Shenzhen, Guangdong 518055, China. [6]Shanghai Synchrotron Radiation Facility (SSRF), Zhangjiang Lab, Shanghai Advanced Research Institute, Chinese Academy of Sciences, 239 Zhangheng Road, Shanghai 201204, China. [7]Instrumentation and Service Center for Molecular Sciences, Westlake University, 18 Shilongshan Road, Hangzhou 310024 Zhejiang Province, China. [8]Shangyu Institute of Semiconductor Materials, Shaoxing 312300, China. [9]Division of Solar Energy Conversion and Catalysis at Westlake University, Zhejiang Baima Lake Laboratory Co., Ltd, Hangzhou, China. [10]These authors contributed equally: Pengju Shi, Jiazhe Xu, Ilhan Yavuz. ✉e-mail: jjxue@zju.edu.cn; yangy@ucla.edu; wangrui@westlake.edu.cn

Secondly, a thicker film with a smaller surface-to-volume ratio tend to have higher resistance to environmental stimuli such as moisture and oxygen[12,13]. As the moisture- or oxygen-induced degradation usually initiates from surfaces or interfaces, reducing the surface-to-volume ratio improves the chemical robustness. Thirdly, a thick layer is beneficial to scalable fabrication by reducing the edge effect and interfacial tension. The increase in film thickness promotes the formation of uniform films with full coverage in large-scale coatings[4,14]. Moreover, a thick-film perovskite layer also helps with device reproducibility[11], which enhances production reliability, a key factor for the industrial competitiveness. However, the PCEs of thick-film PSCs still lag behind those with sub-micron film thickness[15,16]. With the underlying mechanism remaining elusive, the perovskite community has long been under the impression that the limiting factor lies in the short carrier lifetime as a result of defect traps in the film[7,16,17]. Therefore, the photogenerated carriers in a thick-film perovskite system fail to travel as long as to be extracted by the selective contacts. In this study, we constructed a perovskite platform with sufficiently long carrier lifetime to enable carrier extraction by the transporting layers. By utilizing this platform, we found that the internal strain within the perovskite thick films played a crucial role in affecting the device performance. We designed a strain regulation strategy that allowed for the strain retention in the thick perovskite films. A greatly improved PCE from 17.0% to 23.5% (comparable to its thin-film counterpart) was thus obtained in thick-film PSCs. This work elucidated the underlying mechanisms responsible for the current low PCEs of thick-film PSCs, by highlighting the largely ignored strain regulation in a thick film, paving the way for the commercialization of PSCs.

## Results

### Rule out the restrictions of carrier lifetime

We first successfully established a perovskite system with an extraordinarily long carrier lifetime by introducing benzamidine hydrochloride (BZM) with an optimized concentration (Supplementary Fig. 1) into the perovskite precursor (Fig. 1a). Time resolved photoluminescence (TRPL) measurements revealed a typical carrier lifetime of 15.7 μs. For independent verification of such long carrier lifetime, we sent out the perovskite sample to a third-party laboratory for additional TRPL measurements, which gave a consistent result of 15.9 μs (Supplementary Fig. 2). The long carrier lifetime can be attributed to the substantially improved film morphology and crystallinity (Supplementary Fig. 3) due to the retarded the crystallization of the perovskite (Supplementary Fig. 4). Fourier Transform infrared spectroscopy (FTIR) measurements revealed interaction between the BZM and the $PbI_2$, which made the FAI have to compete with BZM to interact with the Pb−I, thus resulting in the sluggish crystallization kinetics of perovskites (Supplementary Fig. 5). Moreover, the introduction of BZM led to the formation of low-dimensional perovskite phases (Supplementary Fig. 6). These low-dimensional perovskite phases could effectively passivate defects in the perovskite film when residing at the grain boundaries as evidenced by the enhanced PL intensity (Supplementary Fig. 7) which suggested reduced nonradiative recombination of charge carriers. Such long carrier lifetimes granted us an opportunity to make thicker perovskite films with sufficiently long charge carrier diffusion lengths to enable efficient charge collection[3]. To systematically investigate the effects of film thickness on the optoelectronic properties of the films, we varied the thickness of the perovskite films by varying the concentration of $PbI_2$ to be 1.4, 1.5, 1.6, 1.8, and 2.0 M and the concentration of FAI/MACl proportionally adjusted accordingly. For convenience, the precursor concentrations, i.e., 1.4 M, 1.5 M, 1.6 M, 1.8 M, and 2.0 M, are used to represent the film thicknesses in this paper.

The carrier lifetime from both the top and bottom regions of these perovskite thin films was investigated by further TRPL measurements from either side of the samples. Figure 1b shows the statistics of the fitted TRPL results collected from 50 films with different

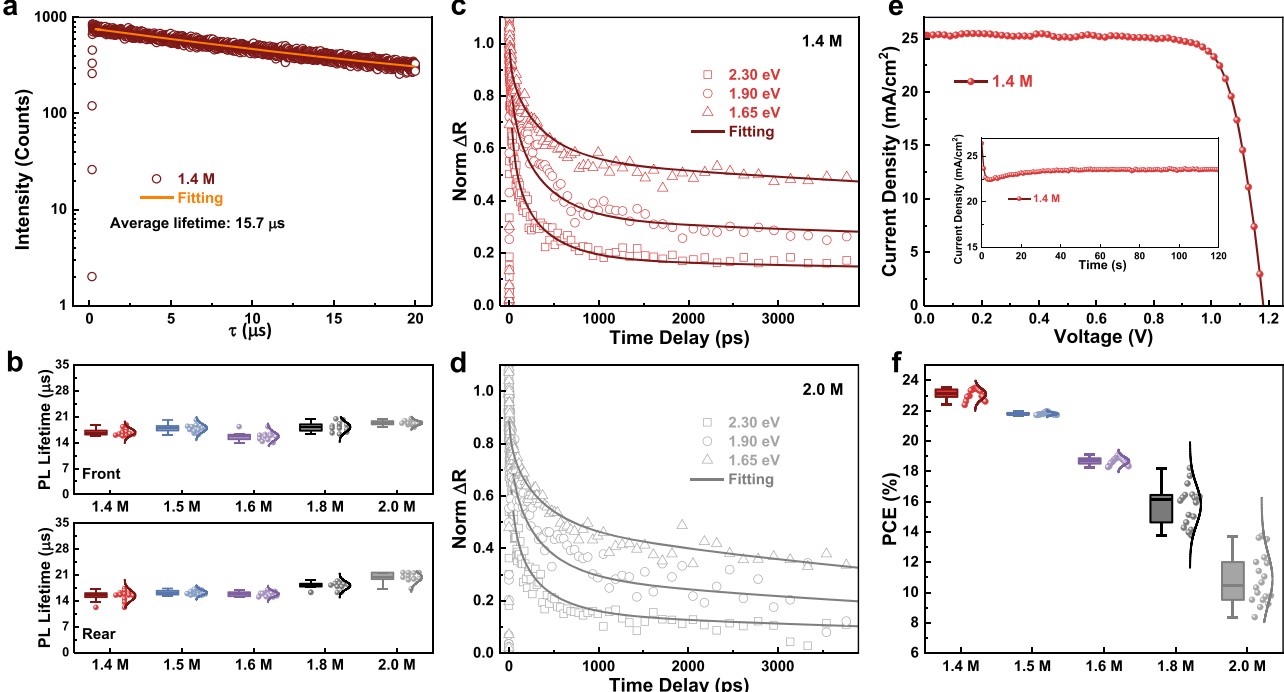

**Fig. 1 | Construction of a perovskite platform with long carrier lifetime. a** Time-resolved photoluminescence spectrum of the perovskite film with BZM. **b** Statistics of the carrier lifetimes measured by TRPL from both the rear and front side of the perovskite films with different thicknesses. Surface-carrier kinetics pumped at 2.30, 1.90, and 1.65 eV probed by transient optical reflection spectra and fitted with a diffusion model for the 1.4 M (**c**) and the 2.0 M (**d**) perovskite. **e** Power conversion efficiency of the PSC for the 1.4 M perovskite with the steady output at a bias of 0.95 V inserted. **f** Statistics of PCEs collected from 100 PSCs with different film thicknesses. For the statistical analysis, center line, median; box limits, 25th and 75th percentiles; curve, normal distribution curve; whiskers, outliers.

thicknesses (Supplementary Figs. 8–19). All spectra were fitted with the single-exponential function and the average lifetime data are summarized in Supplementary Table 1 (rear side) and Table 2 (front side). Negligible variation in the carrier lifetime was observed for the perovskite samples with various film thicknesses, which was in line with carrier lifetime extracted from the transient photovoltage decay measurements (Supplementary Fig. 20). Measured from either side of the samples, all films had similar carrier lifetimes, indicating the consistency between the top and bottom regions of the films. The charge carrier kinetics was further probed by transient reflection spectra (TRS) for the two most representative cases, i.e., the 1.4 M perovskite with the smallest film thickness (Fig. 1c), and the 2.0 M perovskite with the largest film thickness (Fig. 1d). The diffusion coefficient was extracted from the TRS measurements under a pump energy of 2.30, 1.90, and 1.65 eV. The 1.4 M and 2.0 M perovskites showed diffusion coefficients of 0.97 and 1.01 cm$^2$/s respectively (see Supplementary Note 1 for modeling details)[18]. Combining the TRPL and TRS results, the diffusion length was calculated to be 4.04 μm for the 1.4 M perovskite and 4.44 μm for the 2.0 M perovskite. This further confirms that the diffusion length is long enough to enable efficient charge transport and collection.

We subsequently evaluated the photovoltaic performance of the perovskite films with different thicknesses. PSC devices based on the 1.4 M perovskite yielded an average PCE of 23.1% under reverse scan with a stabilized power output of 22.3% as shown in Fig. 1e (best PCE 23.5%, Supplementary Table 3). However, the PCE successively decreased as the film thickness increased. As shown in Fig. 1f and Supplementary Figs. 21 and 22), the PCE dramatically dropped from an average PCE of 23.1% for the 1.4 M perovskite to an average PCE of merely 10.9% for the 2.0 M perovskite. The PCE decrease for the thicker films mainly originated from the dramatically reduced fill factor (from 0.77 to 0.45) as a result of the significantly increased series resistance (from 38.5 Ω to 116.0 Ω, Supplementary Table 4). Despite the long diffusion length of the 2.0 M perovskite, the device PCE is significantly lower than that of the 1.4 M perovskite, indicating the existence of separate factors, other than carrier diffusion length, that limited the PCE of thick-film PSCs.

**Critical role of lattice strain in thick perovskite films**

We first examined the film chemistry to rule out any influence from composition variation. X-ray photoelectron spectroscopy (XPS) of the perovskite films showed no obvious difference as the film thickness increased (Supplementary Fig. 23). We also performed time-of-flight secondary ion mass spectrometry (SIMS) to study the distribution of chemical compositions throughout the film. The distribution profiles of $Pb^{2+}$, $Cs^+$, $FA^+$, and $In^+$ (as a reference to indicate the interface) were displayed in Supplementary Fig. 24. The perovskite thick film showed a very similar composition distribution to that of its thin film counterpart. These results ruled out the effect of the compositional distribution on the device performances.

As the fill factor and the series resistance in PSCs are highly related to the conductivity of the perovskite film, we investigated the conductivity dependence on the film thickness. Figure 2a shows statistics of the conductivity of the perovskite films collected by conductive atomic force microscopy (Supplementary Fig. 25), with the film thickness measured by a step profiler (Supplementary Fig. 8). The electrical conductivity of the perovskite film reduced as the film thickness increased. The conductivity dropped from $5.5 \times 10^{-3}$ S/m in the 1.4 M perovskite to $2.5 \times 10^{-3}$ S/m in the 2.0 M perovskite, which led to an increased series resistance and thus reduced PCE in thick-film PSCs. To obtain further insights into the underlying factors that affected the conductivity of the films, we studied the influence of lattice structures via X-ray diffraction (XRD, Supplementary Fig. 26). All samples exhibited two dominant diffraction peaks at around 14° and 28°, which can be assigned to the characteristic (001) and (002) planes

of the α-phase perovskite. Notably, the (001) diffraction peak gradually shifted from -14.07° to -14.01° as the film thickness increased from 1.4 M to 2.0 M (Fig. 2b). The peak shift to lower diffraction angle indicated an expansion of the perovskite lattice with increasing thickness, which could be attributed to the strain effect induced by the substrate. In a relatively thinner perovskite film, the surface tension may impose greater strain effects that lead to the lattice compression. As the film thickness increases, strain relaxation results in the recovery of the lattice. The strain distribution within the film was further investigated through depth-resolved grazing-incidence wide-angle scattering (GIWAXS) by varying the incident beam angle from 0.3° to 1.0° with a step increase of 0.1° (see detailed information in Supplementary Note 2 and Supplementary Fig. 27). As the incident angle varied from 1.0° to 0.5°, the diffraction patterns of the 1.4 M perovskite gradually shifted downwards from a $Q_z$ value of 0.980 Å$^{-1}$ to 0.975 Å$^{-1}$ (Fig. 2c), suggesting a more pronounced compressive strain around the region closer to the substrate. In contrast, the strain effect vanished in the case of thicker films. Figure 2d shows the diffraction patterns of the 2.0 M perovskite. The peak position at -0.975 Å$^{-1}$ was invariant as the incident angle ranged from 1.0° to 0.5°, suggesting that the lattice strain relaxed in the thicker film and became independent of the depth of the film. Figure 2e depicts the effect of film thickness on the lattice strain of perovskites, demonstrating the presence of lattice compression in thinner films, but strain relaxation in thicker films.

We subsequently investigated how lattice strain could have affected the conductivity of the perovskite films. Since conductivity is governed by carrier mobility and carrier concentration, we investigated these two factors for the 1.4 M and 2.0 M perovskite films. The space-charge-limited current technique was employed to measure carrier mobility. The electron and hole mobilities were obtained by constructing electron- or hole-only devices (Fig. 3a, b, Supplementary Fig. 28 and Supplementary Tables 5–7, see Supplementary Note 3 for details of the calculation method). We observed a gradual increase in the electron mobility from -10 to -100 cm$^2$ V$^{-1}$ s$^{-1}$, whereas the hole mobility remained almost invariant as the film becomes thicker. Given the negligible changes in the carrier lifetime, the mobility increase could be ascribed to the change in effective mass as a result of the lattice strain. In light of this, we investigated the band dispersion of perovskites under different strain modes by first-principles calculations (Supplementary Figs. 29, 30). Figure 3c, d shows the band dispersion of the valence band maximum (VBM) and conduction band minimum (CBM), respectively, of perovskites under different strain modes (Supplementary Note 4). The results reveal a less dispersive CBM structure for the 1.4 M perovskite under compressive strain, indicating an increased electron effective mass of 0.103 $m_0$, as compared to that of 0.075 $m_0$ in the strain relaxed 2.0 M perovskite (Supplementary Table 8). The VBM band dispersion showed negligible difference between the 1.4 M and 2.0 M perovskites, suggesting a similar hole effective mass for the two cases. For the thicker perovskite film, a reduced electron effective mass but unaffected hole effective mass is consistent with the trends measured for the electron and hole mobilities, verifying the important role of strain in affecting the perovskite carrier mobility.

Given the opposite trend in the change of carrier mobility to that in conductivity, we further evaluated the carrier concentration in the thick perovskite films. The carrier concentration of the perovskite films can be derived from the slope of the capacitance-voltage (Mott-Schottky) plots (see the measurement and calculation details in Supplementary Fig. 31, Supplementary Table 9 and Supplementary Note 5). The electron concentration (Fig. 3e) decreased from -1.6 × 10$^{16}$ to -0.5 × 10$^{16}$ cm$^{-3}$ with increasing film thickness. Drive-level capacitance profiling also revealed a gradual decrease in carrier concentration as the perovskite film became thicker, further verifying the change in carrier concentration (Supplementary Fig. 32, Supplementary Note 6 and Supplementary Fig. 33). Two possible mechanisms could

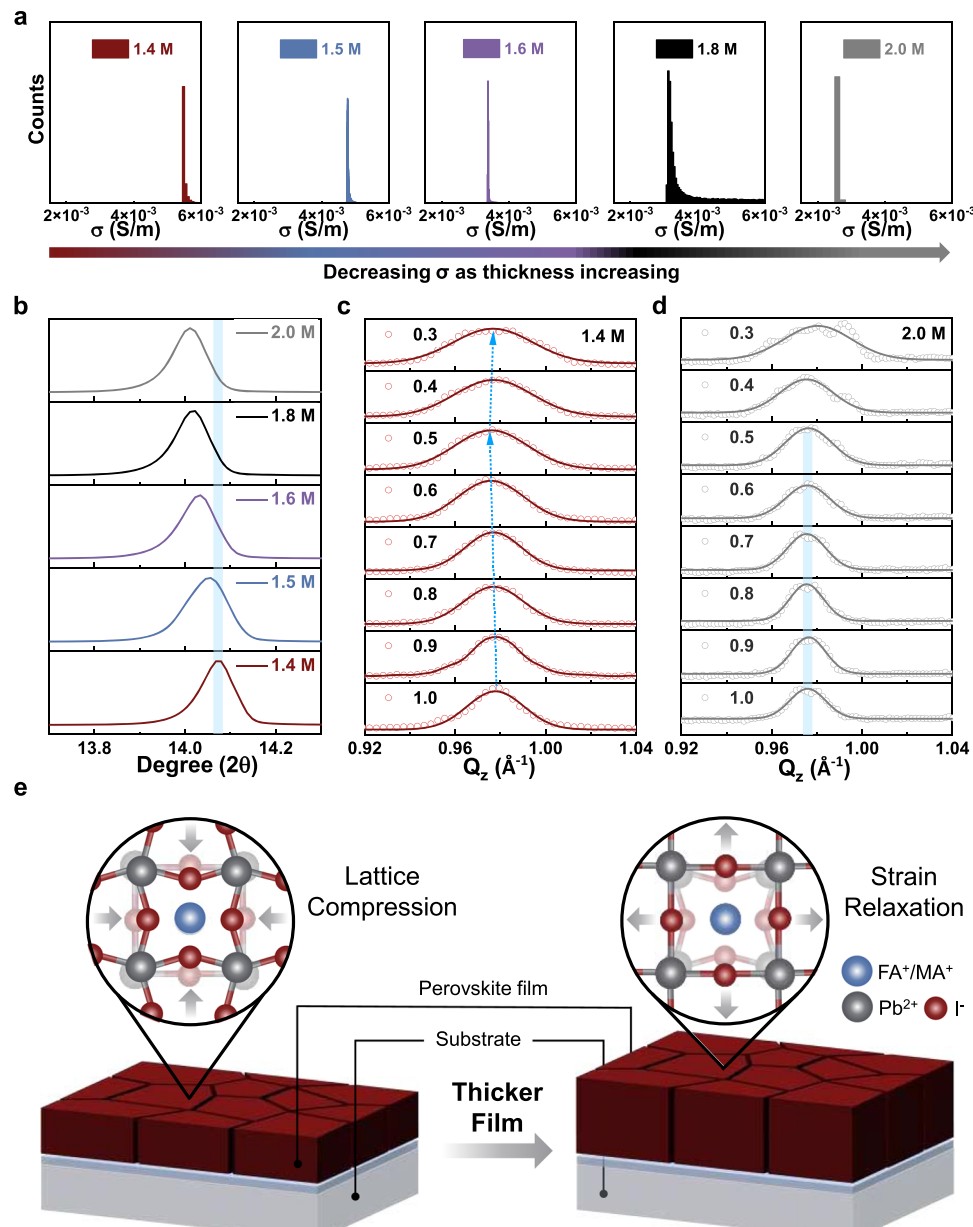

**Fig. 2 | Investigations on the lattice strain in perovskite films with various thickness. a** The statistics of the conductivity of perovskite films collected by the conductive atomic force microscope. **b** X-ray diffraction patterns of perovskite films with different thicknesses. Grazing-incident wide-angle X-ray scattering profiles of 1.4 M (**c**) and 2.0 M (**d**) perovskite films with the angle of incident beam ranging from 0.3° to 1.0°. **e** Schematics of the lattice strain in perovskite films with different thicknesses.

contribute to the carrier concentration change. On one hand, the compressive strain induced lattice distortion which led to the changes in effective mass as demonstrated above. As the carrier concentration is proportional to the effective mass, the carrier concentration would decrease for thicker perovskite films as a result of the decreased effective mass[19,20]. On the other hand, the perovskite carrier concentration could also be affected by unintentional doping by shallow-level defects[21,22]. Compressive strain would strengthen the chemical bonds and reduce the defect formation energies of some shallow-level defects[23] (Supplementary Note 7 and Supplementary Fig. 34). Therefore, though a thicker perovskite film exhibited a higher mobility, the low conductivity limited its PCE as a result of the strain-induced decrease in electron concentration. The strain-induced lattice distortion led to an increase in electron mobility and a decrease in electron concentration. The reduced electron concentration finally resulted in a

decrease in the conductivity of the thicker perovskite films, and thus limited their photovoltaic performance. Moreover, the strain variation occurs closer to the bottom region of the films, which is towards the electron transporting layer. Therefore, the strain-induced variation in the electron behavior may have a more pronounced effect on the photovoltaic performance, which also in turn verifies the change in the electron concentration and electron mobility.

To further confirm that the strain effect is a critical factor limiting the PCE of thick-film PSCs, a strain regulation strategy (SRS) was designed to mitigate the strain relaxation for the thick perovskite films. In light of the interface-induced compressive strain, the SRS we devised involved a layer-by-layer deposition of perovskite films which introduced additional interfaces for strain-inducing. The PbI2 solution was first deposited onto the substrate, followed by the sequential deposition of the precursor solutions containing the organic cations.

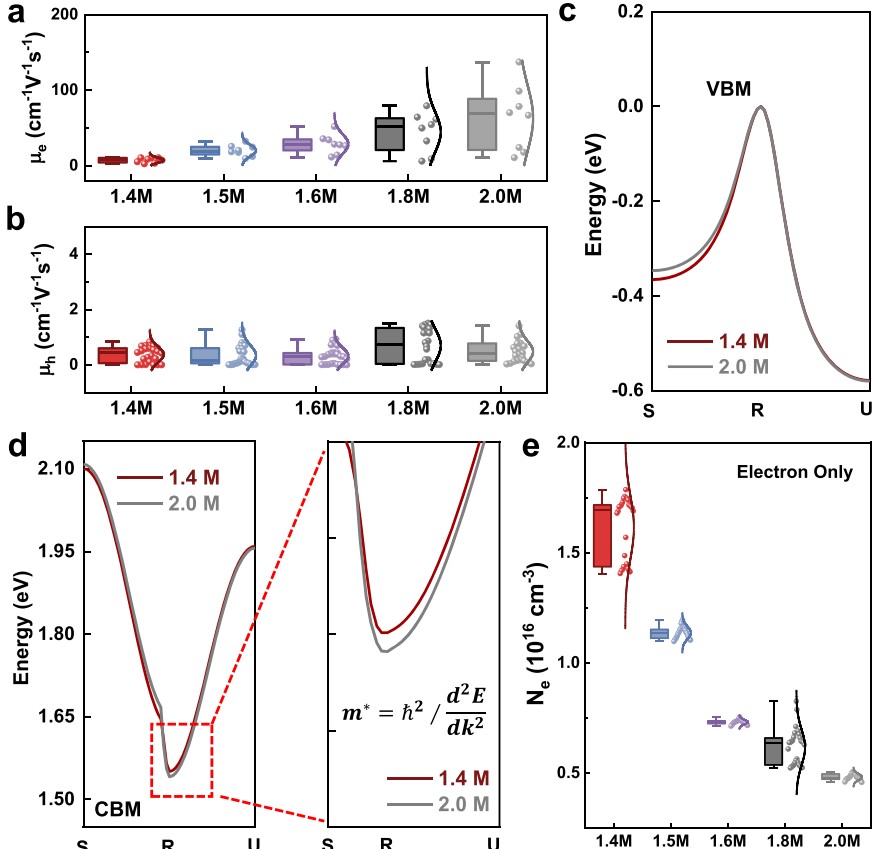

**Fig. 3 | Electronic properties of perovskite films under different strain modes. a, b** Statistics of electron and hole mobility extracted from space-charge-limited-current (SCLC) measurement. Calculated band structures based on first-principal density functional theory for valence band maximum (**c**) and conduction band minimum (**d**). **e** Statistics of electron concentration derived from capacitance-voltage (Mott-Schottky) plot. For the statistical analysis, center line, median; box limits, 25th and 75th percentiles; curve, normal distribution curve; whiskers, outliers.

The first round of the precursor dripping led to the formation of a thin perovskite intermediate layer through the reaction between the pre-deposited $PbI_2$ and the precursor solution containing organic cations (Supplementary Figs. 35, 36). XRD measurements revealed that this thin intermediate perovskite layer exhibited the same compressive lattice strain as the case of 1.4 M (Supplementary Fig. 37). This layer can thus serve as a foundation for maintaining strain continuity during the subsequent second-round deposition of the perovskite precursor. This approach ensured that the compressive strain was maintained within each thin perovskite layer, which can be seen as a "quasi substrate" to induce lattice strain, and finally led to the formation of a thick perovskite layer with compressive strain in a manner akin to epitaxial growth. XRD analysis in Fig. 4a, b verified the compressive strain retention for the perovskite films fabricated with SRS, which is observed by the similar peak position to that of the 1.4 M perovskite. Figure 4c shows the strain distribution of the thick-film perovskite with SRS as revealed by depth-resolved GIWAXS. Mimicking the case of the 1.4 M perovskite, the diffraction peak shifted from a $Q_z$ value of 0.980 Å to 0.975 Å as the incident beam angle ranges from 1.0° to 0.7°, suggesting the existence of compressive strain around the bottom region of the film (see the schematic illustration in Supplementary Fig. 38). As SRS successfully allowed retention of the compressive strain in the thick perovskite film, the electron concentration increased (Supplementary Figs. 39–45). As a result, its conductivity enhanced from ~0.0025 to ~0.0060 S/cm (Fig. 4d and Supplementary Fig. 46). The PCE based on SRS improved from 10.9% (best 17.0%) for PSCs based on the 2.0 M perovskite to 22.9% (best 23.5%, certified 23.1%) (Fig. 4e and Supplementary Figs. 47, 48). The fill factor improved from 0.45 to 0.75, with a reduced series resistance from 116.0 Ω to 38.5 Ω

(Supplementary Fig. 49). The success of our SRS strategy further verifies the significance of compressive strain in constructing thick-film PSCs.

## Discussion

In this work, we unveiled the underlying mechanisms for the low PCEs of thick-film PSCs, providing mechanistic insights into the limiting factors. We first established a perovskite system with sufficiently long carrier lifetime, which allowed us to preclude the restrictions of carrier lifetime on the device performance. Subsequently, based on this perovskite platform, we demonstrated that lattice strain and thus the change of carrier concentration is responsible for the PCE variations in thick-film perovskite photovoltaic devices. We further showed that the lowered PCE in thick-film PSCs can be recovered using SRS, verifying the significant role of strain regulation in realizing high-performance thick-film PSCs.

## Methods
### Materials
Unless stated otherwise, solvents and chemicals were obtained commercially and used without further purification. N, N-dimethylformamide (DMF) (anhydrous, 99.8%), dimethyl sulfoxide (DMSO) (anhydrous, ≥99.9%), chlorobenzene (CB) (anhydrous, 99.8%), isopropanol (IPA) (anhydrous, 99.5%), water (ACS reagent), t-BP (99%), Li-TFSI (99.95% trace metals basis), $PbI_2$ (99.999%, perovskite grade), Cesium Iodide (CsI, 99.999%), silver (Ag) and gold (Au) were obtained from Sigma-Aldrich Inc. MACl (99%) and FAI was obtained from Great Cell. C60, PCBM, PEDOT:PSS, FK209 and Spiro-OMeTAD (99.8%) were obtained from Xi'an Polymer Light Technology Corp.

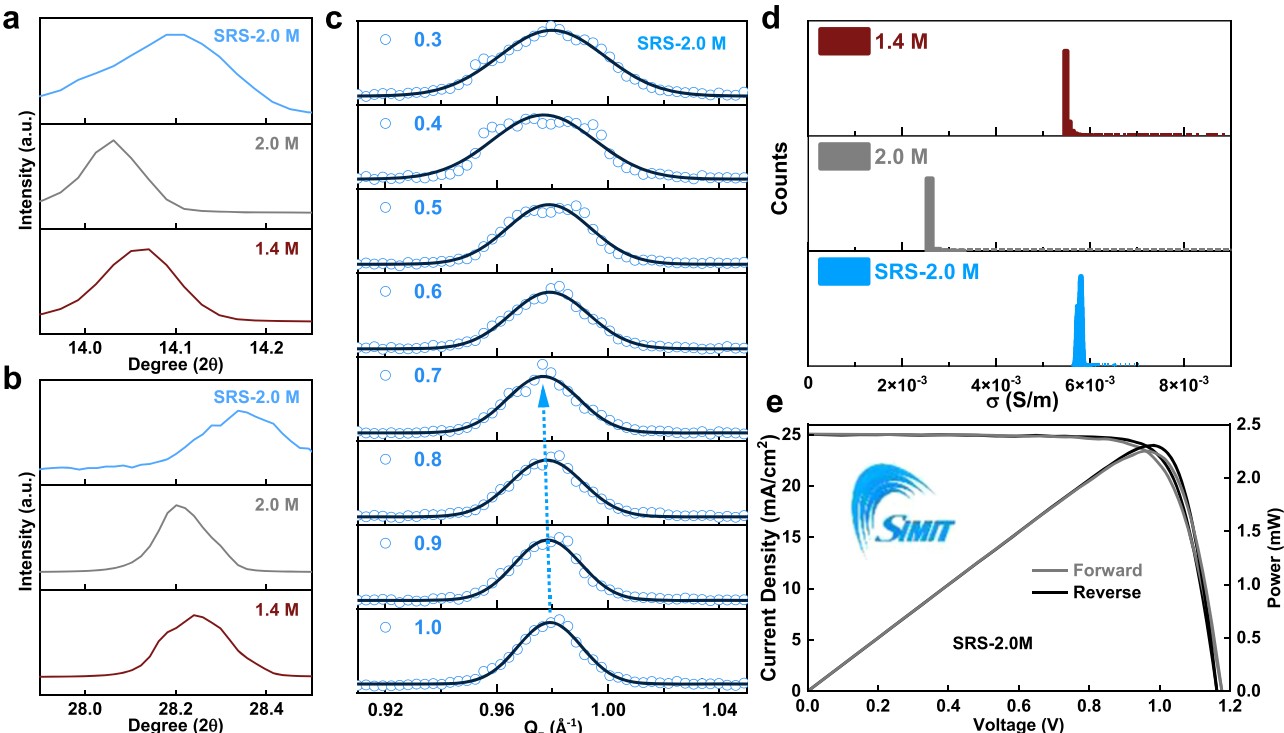

**Fig. 4 | Strain regulation in thick-film PSCs.** Comparison of the XRD patterns of 1.4 M perovskite (**a**) and 2.0 M (**b**) perovskite, and perovskite thick film deposited via strain regulation strategy (SRS). **c** GIWAXS profiles of the thick perovskite film deposited via SRS with the angle of incident beam angle ranging from 0.3° to 1.0°.

**d** Comparison of the conductivity of SRS, 2.0 M and 1.4 M perovskite films.
**e** Certified *J-V* curves of SRS device with the logo of SIMIT inserted. Permission to use the logo was obtained from SIMIT.

Tin Oxide ($SnO_2$) nanoparticle (15 wt% in water) was obtained from Alfa-Aesar Inc. Benzamidine hydrochloride (98%) were purchased from Aladdin. Hydriodic acid (HI, 57wt%) was purchased from TCI. Hypo phosphorous ($H_3PO_2$) was purchased from J&K Scientific.

## Device fabrication

Perovskite solar cells were fabricated with the following structure: indium tin oxide (ITO)/$SnO_2$/$FA_xCs_{1-x}PbI_3$/Spiro-OMeTAD/Ag or Au. The ITO glass was pre-cleaned in an ultrasonic bath of acetone and isopropanol and treated in ultraviolet ozone for 20 min before use. A thin layer (ca. 30 nm) of $SnO_2$ was spin-coated onto the ITO glass and baked at 165 °C for 35 min. $SnO_2$ solution was diluted in water (2 mg mL$^{-1}$) before spin-coating. After cooling to room temperature, the glass/ITO/$SnO_2$ substrates were transferred into a nitrogen glove box. The $PbI_2$ solution was prepared by dissolving 1.4 mM $PbI_2$ and 0.07 mM CsI into 1 mL DMF/DMSO mixed solvent (v/v 94/6). The FAI/MACl solution for the REF was prepared by dissolving 80 mg FAI, 13 mg MACl into 1 mL IPA. The FAI/MACl/Benzamidine hydrochloride (BZM) (1.4 M) solution was prepared by dissolving 80 mg FAI, 13 mg MACl and 4 mg BZM into 1 mL IPA. The solutions should be stirred overnight before use. To fabricate the perovskite layer of 1.4 M, 1.5 M, 1.6 M, 1.8 M and 2.0 M, the $PbI_2$ solution was spin-coated on the substrate at 1500 rpm for 40 s, and then, the FAI/MACl/BZM solution was spin-coated on the $PbI_2$ film at 1800 rpm for 40 s, followed by pre-annealing inside the glove box at 90 °C for 1 min and annealing outside the glove box at 150 °C for 10 min with 40% humidity. The Spiro-OMeTAD solution [60 mg Spiro-OMeTAD in 700 μL CB with 25.5 μL t-BP, 15.5 μL Li-TFSI (520 mg/mL in ACN) and 12.5 μL FK209 (375 mg/mL in ACN)], was spun onto the perovskite film as a hole conductor. The devices were completed by evaporating 100 nm gold or silver in a vacuum chamber (base pressure, $5 \times 10^{-4}$ Pa). The aperture area of the device is 0.1 cm², designated by the shadow mask. For the perovskite layer

fabricated by SRS, the $PbI_2$ solution was spin-coated on the substrate at 1500 rpm for 40 s, and then, the FAI/MACl solution (FAI/MACl 80 mg/13 mg in 1.7 mL IPA) was spin-coated on the $PbI_2$ film at 1800 rpm with sequential dripping of FAI/MACl/BZM (FAI/MACl/BZM 80 mg/13 mg/4 mg in 1 mL IPA) solution at 1800 rpm for 180 s. The sequential dripping of the FAI/MACl is the key to enabling the fabrication of a perovskite film with a thickness over 2 μm while retaining the compressive strain in the film. After that, the films were pre-annealed inside the glove box at 90 °C for 1 min and annealed outside the glove box at 150 °C for 10 min with 40% humidity. For perovskite layer of 1.5 M, 1.6 M, 1.8 M and 2.0 M, the concentration of $PbI_2$ and CsI increased accordingly with invariant amount of DMSO and DMF, and the amount of FAI/MACl/BZM is 90/14.6/4.5 mg, 100/16.2/5 mg, 120/19.5/6 mg and 140/22.75/7 mg in 1 mL IPA respectively. For the fabrication of electron-only devices, PCBM layer was deposited at 2000 r/min for 30 s by dissolving 15 mg PCBM and 5 mg C60 in 1 mL CB. For the fabrication of hole-only devices, PEDOT:PSS layer was deposited by directly spinning the purchased PEDOT:PSS at 4000 r/min for 30 s, followed by annealing at 165 °C for 35 min. The wide-bandgap perovskite films were fabricated with the composition of $Cs_{0.05}FA_{0.8}MA_{0.15}PbI_{2.25}Br_{0.75}$.

## Device characterization

*J-V* characteristics of photovoltaic cells were taken using a Keithley 2400 source measure unit under a simulated AM 1.5 G spectrum, with an Oriel 9600 solar simulator (Enlitech). Typically, the devices were measured in reverse scan (1.25 V ⟶ 0 V, step 0.02 V). All the devices were measured without pre-conditioning such as light-soaking and applied a bias voltage. Steady-state power conversion efficiency was calculated by measuring stabilized photocurrent density under a constant bias voltage. External quantum efficiencies were measured using an integrated system (Enlitech) and a lock-in amplifier with a current preamplifier under short-circuits' condition.

## Materials characterization and spectroscopic investigation

UV-vis absorption spectra of the perovskite films were obtained using a Shimadzu UV-VIS-NIR (UV3600Plus + UV2700) equipped with integrating sphere, in which monochromatic light was incident to the substrate side. XPS measurements were carried out on an XPS (ThermoFisher ESCALAB Xi+). An Al Kα (1,486.6 eV) X-ray was used as the excitation source. Briefly, the TR measurements were performed by a pump–probe spectrometer (TA-100, Time-tech spectra). A Ti:Sapphire amplifier (Astrella, Coherent) is used to generate 800 nm light at 1 kHz repetition rate. The fundamental pulse is split into two parts. One part is sent to an optical parametric amplifier (TOPAS, Lightconversion) for the various pump wavelength generation. The pump is chopped at a frequency of 500 Hz and attenuated by neutral density filter wheels. The other part of the fundamental pulse is focused into a sapphire crystal to generate a visible continuum (450–810 nm) that is used as the probe. The probe pulses are delayed in time with respect to the pump pulses using a motorized translation stage mounted with a retroreflecting mirror. The pump and probe are spatially overlapped on the surface of the sample. The incident angle for pump is around 0° and probe is around 45°. Conductive atomic force microscope was measured with the machine (Cypher ES Oxford Instruments) by a bias of 500 mV. For TRPL measurement, the sample was excited with a picosecond pulsed diode laser (Pico-quant LDH 450), with a ~70 ps pulse width and 20 MHz repetition rate, focused on sample with a 100× objective (NA = 0.90). The PL signal was acquired through the TCSPC strobelock system. The total instrument response function for the PL decay was less than 200 ps, and the temporal resolution was less than 30 ps. The energy density of laser for TRPL measurements was 26.46 nJ/cm$^2$. The thickness of perovskite films was measured with a Surface Profile Measuring System (DektakXT). XRD experiments were performed on sealed-tube Cu X-ray source, equipped with 1D LynxEye detector. The resolution of the XRD measurement is 0.0001°. Prior to XRD analysis, all samples were positioned at the same height, aligned with the edge of the holder's top surface. The samples were subjected to a spin rate of 20 r/min to ensure averaged results. SEM was carried out on the Field Emission Environment Scanning Electron Microscope of Quattro S. For characterization of strain, carrier mobility and carrier concentration, the details are seen corresponding in Supplementary Texts. Time of Flight secondary ion mass spectrometry (ToF-SIMS) depth-profile analysis was performed using a PHI NanoTOF III instrument (ULVAC-PHI, Inc.), where a 3 kV Ar ion beam was used for erosion and a 25 keV Bi$^+$ pulsed primary ion beam was used for the analysis. The area of analysis was $100 \times 100\ \mu m^2$ while the sputtering area was $400 \times 400\ \mu m^2$. The single crystal was grown via the inverse temperature crystallization method. 0.64 mmol BZM, and 1.04 mmol PbI$_2$ were added into 3 mL hydriodic acid (HI, 57wt%, TCI) and 0.1 mL hypo phosphorous acid (H$_3$PO$_2$, J&K Scientific). After heating and complete dissolution, the solution was put into a muffle furnace for cooling and high-purity halide perovskite single crystal was obtained after around 48 h. FTIR spectroscopy was obtained using FT/IR-6100 (Jasco).

### Reporting summary

Further information on research design is available in the Nature Portfolio Reporting Summary linked to this article.

## Data availability

Crystallographic data for the single crystal reported in this article have been deposited at the Cambridge Crystallographic Data Center (CCDC), with deposition number 2334815 corresponding to (BZM)$_2$Pb$_3$I$_8$. These crystallographic data can be obtained free of charge via https://www.ccdc.cam.ac.uk/structures/ with the deposition number (2334815). The main data generated in this study are provided in the Supplementary Information/Source Data file. All other data supporting the findings of this study are available from the corresponding authors on request. Source data are provided with this paper.

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

## Acknowledgements

All the authors thank Dr. Shuai Yuan and Prof. Xiangyang Ma for the helpful discussion during this project, Mr. Ming Xia from School of Engineering in Westlake University for the assistance in single-crystal analysis, Dr. Chenhui Zhu from Beamline 7.3.3, Advanced

Light Source, Lawrence Berkeley Laboratory for the assistance in GIWAXS measurement, Prof. Lei Meng from ICCAS for the assistance in TRPL measurement, Dr. Yinjuan Chen from Instrumentation and Service Center for Molecular Sciences at Westlake University for the assistance in the characterizations, and Dr. Xiaohe Miao, Dr. Lin Liu, Dr. Jiaqi Guan, Dr. Chao Zhang, Dr. Pei Sheng from Instrumentation and Service Center for Physical Sciences (ISCPS) at Westlake University for the assistance in the characterizations. J.X. and R.W. acknowledge the grant (LD22E020002, LD24E020001) by Natural Science Foundation of Zhejiang Province of China. J.X. acknowledges the financial support by Shanxi-Zheda Institute of Advanced Materials and Chemical Engineering (2021SZ-FR006). J.X. acknowledges the grant by the National Natural Science Foundation of China (grant no. 62274146) and the grant (LR24F040001) by Natural Science Foundation of Zhejiang Province of China. R.W. acknowledges the funding from State Key Laboratory of Silicon Materials and Advanced Semiconductor Materials (SKL2022-07) and the support of Key R&D Program of Zhejiang Province (2024SSYS0061).

## Author contributions

P.S., J. Xue. and R.W. conceived the idea and designed the experiments. P.S. conducted the experiments under the supervision of J.X. and R.W. P.S. wrote the first draft and J.X. wrote the final manuscript. I.Y. performed the DFT calculations. T.H., S.T. carried out the GIWAXS for detailed strain analysis at beamline 7.3.3 at LBL under the supervision of Y.Y. Y.L. helped to grow the $(BZM)_2Pb_3I_8$ crystal under the supervision of E.S. J.Xu., S.W., Z.C., W.F., X.Z., Y.T., K.Z., C.W., X.G. and D.J. assisted to prepare samples and perform the characterizations. X.C. and X.Y. performed the TRS investigations. D.Y. and Y.Y. helped to modify the manuscript.

## Competing interests

The authors declare no competing interests.
