## [Peer Review File · Nature Communications]

Strain Regulates the Photovoltaic Performance of Thick-Film PerovskitesREVIEWER COMMENTS

Reviewer #1 (Remarks to the Author):

In this work, Shi et al. presented a perovskite platform exhibiting a long carrier lifetime and revealed the pivotal role of internal strain within the thick perovskite films in influencing the device performance. The authors further designed a strain regulation strategy that allowed for the strain retention in the thick perovskite films. A greatly improved PCE from 17.0% to 23.5% was obtained in thick-film PSCs. Overall this manuscript is very well-written, and the results are interesting. However, there are certain issues that need clearer explanations and appropriate addressing before the manuscript can be considered for publication. Detailed comments are listed below.

- 1) The authors attribute the long carrier lifetime to the substantially improved film morphology and crystallinity with the incorporation of BZM. The main evidence is SEM images. However, it's not clear that why BZM-based film exhibit such a long carrier life time of 15.7 μ s. There are a lot of published works that exhibit high film quality but not such long carrier lifetime. What are the reasons behind the long carrier lifetime? Some long carrier lifetime
- 2) Numerous organic salts have been employed as additives in perovskite films, with similar BZM compounds already having been used in perovskite solar cells. What role does BZM play in the perovskite film? Furthermore, does the concentration of BZM affect the performance of devices?"
- 3) One key focus of this study revolves around the utilization of the SRS method for producing high-quality perovskite films. However, this part lacks in-depth discussions concerning the underlying reasons for this choice. Therefore, it becomes pertinent to delve into the factors that set the SRS method apart and elucidate its differentiating aspects.
- 4) It's interesting that the authors observed the retention of compressive strain in the perovskite films fabricated using SRS. Why does this method make such a substantial difference in thick film devices? It is recommended that the authors provide further discussion, considering that readers would be highly interested in understanding the underlying reasons. Additionally, it is suggested to include supporting data for SRS-based films, such as SEM and XRD."
- 5) The authors use two-step method to fabricate the perovskite films. So, the following description is not clear in the manuscript, "perovskite films were made by using perovskite precursor concentrations of 1.4, 1.5, 1.6, 1.8, and 2.0 M,"

Reviewer #2 (Remarks to the Author):

In this work, the authors have chosen a very interesting topic, which is "strain regulates the photovoltaic performance of thick-film perovskites". They ruled out the restrictions of carrier lifetime on the device

performance by using additive BZM and delved into the critical role of the lattice strain in micron-scale thick perovskite films. Finally, an effective method to regulate strain has been applied to the fabrication of thick-film perovskite. In general, this paper has substantial data and impressive results. Thus, reviewer believe this work deserves being published in Nature Communications after revise.

Some comments are listed below for the further improvement:

1. In figure 2b, the authors present XRD patterns to study the difference of lattice structures. The (001) diffraction peak gradually shifted from $\sim 14.07^\circ$ to $\sim 14.01^\circ$ and this range is very small. And the authors have mentioned that "The peak shift to lower diffraction angles indicated an expansion of the perovskite lattice with increasing thickness, which could be attributed to a strain effect induced by the substrate." What is the substrate of the XRD test samples? Is the XRD pattern of the substrate used for calibration during data processing? The authors described the XRD characterization too simply.
2. This manuscript put forward strain regulation strategy (SRS), which achieved by sequential deposition of FAI/MACl solution on PbI₂ films. In this deposition method, whether the growth of perovskite crystals is controllable? Is the film preparation reproducible? Why can this deposition method implement the SRS strategy?
3. In figure 1f and Supplementary Fig.16, the performance of perovskite solar cells dropped as the increase of perovskite precursor concentration. Meanwhile, the authors described that "Perovskite films with various thicknesses of 0.80, 0.81, 0.87, 1.22, and 1.36 μm were made by using perovskite precursor concentrations of 1.4, 1.5, 1.6, 1.8, and 2.0 M, respectively (Supplementary Fig. 3)." As usually, as the film thickness increases, the absorbance of the film and the J_{sc} increase. This is contrary to Supplementary Fig.16b. Did the author consider this factor?
4. Wrong description: "(FAI/MACl 80 mg/13 mg/4 mg in 1 mL IPA)".

Reviewer #3 (Remarks to the Author):

This work investigates the underlying mechanisms for the low efficiency of thick-film based perovskite solar cells. By incorporating Benzamidinium hydrochloride, the device performance is enhanced, which is attributed to the enhancement of mobility introduced compressive strain. The device performance is excellent, but the mechanisms are not surprising, which is revealed elsewhere, such as Nature 577, 209 , Nature communications 10, 815, and <https://doi.org/10.1093/nsr/nwab047>

Furthermore, when using XRD measurement to evaluate the strain states, it hypothesizes the composition of the film is uniform. However, based on the formula of this mixed perovskite, it is not sure if the compositional profiles over different samples with different thicknesses are the same, or their distributions are even. It is suggested to examine the final compositional information of different films, which is important to verify the conclusion of this work. Some additional comments are listed below for further improvement of this work.

1. Page 6 line 173: In Figure 2b, the description of “the peak shift to lower diffraction angles indicated an expansion of the perovskite lattice” may not be accurate. The peak shift to lower diffraction angles indicated an increased spacing of the crystal plane parallel to the substrate, how about the lattice spacing perpendicular to the substrate?
2. Perovskite films' strain state (tension or compressive) should be checked by referencing the strain-free state (freestanding perovskite powder scraped from the as-prepared perovskite films from the substrates) or with “ $[\sin]^2 \psi$ method”. The GIWAXS patterns are suggested to be added to the main text or Supplementary Materials, which could be in favor of the analysis of the strain of perovskite film according to the Q_z value of different azimuths.
3. The examination of the film chemistry to rule out any influence from composition variation in thin and thick perovskite films should be more careful. Depth-resolved X-ray photoelectron spectroscopy (XPS) and/or other characterization are suggested to conform to the composition variation.
4. It is interesting to see the significant effect of strain regulation on electron mobility and electron concentration in perovskite. Does this strategy have broader implications, such as how it works with different components?

RESPONSE LETTER (NCOMMS-23-30941)

Reviewer #1 (Remarks to the Author):

In this work, Shi et al. presented a perovskite platform exhibiting a long carrier lifetime and revealed the pivotal role of internal strain within the thick perovskite films in influencing the device performance. The authors further designed a strain regulation strategy that allowed for the strain retention in the thick perovskite films. A greatly improved PCE from 17.0% to 23.5% was obtained in thick-film PSCs. Overall this manuscript is very well-written, and the results are interesting. However, there are certain issues that need clearer explanations and appropriate addressing before the manuscript can be considered for publication. Detailed comments are listed below.

Reply: We appreciate the reviewer's positive comments on this work. We have provided a point-by-point response to all the comments raised.

1) The authors attribute the long carrier lifetime to the substantially improved film morphology and crystallinity with the incorporation of BZM. The main evidence is SEM images. However, it's not clear that why BZM-based film exhibit such a long carrier life time of 15.7 μ s. There are a lot of published works that exhibit high film quality but not such long carrier lifetime. What are the reasons behind the long carrier lifetime? Some long carrier lifetime.

Reply: We thank the reviewer's valuable comments. We have performed additional experiments and analysis to further elucidate the underlying reasons for the long carrier lifetime of BZM-based perovskite films. We performed FTIR measurements to investigate the interaction of BZM with perovskite precursors. As shown in **Fig. R1**, the C=N stretching vibration, and the N-H stretching vibration of BZM shifted from 1672 to 1663 cm^{-1} , and from 3060 to 3048 cm^{-1} respectively, after mixing with PbI_2 , suggesting the binding of BZM to the Pb^{2+} through the amidinium group. This interaction could slow down the crystallization of the perovskite because the FAI had to compete with BZM to interact with the Pb-I. The slower crystallization kinetics was indicated by the color change of the perovskite during the film fabrication process. The control film swiftly turned dark upon annealing (after ~ 2 seconds), whereas the film with BZM still appeared a bit yellow till 4 seconds (**Fig. R2**). The sluggish crystallization process led to enlarged grain sizes (see the SEM images) and enhanced crystallinity as revealed by the higher diffraction intensity in the XRD measurements. From the XRD patterns of the BZM-based perovskite films, we also found that the introduction of BZM led to the formation of low-dimensional perovskite phases. This was backed by the XRD peaks at low diffraction angles (**Fig. R3**). These low-dimensional perovskite phases could effectively passivate defects in the perovskite film when residing at the grain boundaries as evidenced by the enhanced PL intensity which suggested reduced non-radiative recombination of charge carriers (**Fig. R4**). As a result, the perovskite films with BZM showed prolonged carrier lifetime due to improved film

morphology, enhanced crystallinity, and reduced defect states (**Fig. R5**).

Fig. R1 FTIR spectra of pure BZM and BZM mixed with PbI_2 .

Fig. R2 Photos of the perovskite films with and without BZM during the annealing process.

Fig. R3 Enlarged XRD patterns at low diffraction angles of the 1.4 M perovskite film with BZM.

Fig. R4 PL spectra of 1.4 M perovskite thin films with and without introducing BZM.

Fig. R5 TRPL plots of 1.4 M perovskite with and without introducing BZM.

We included the data in the Supplementary Materials and modified the manuscript as follows:

“The long carrier lifetime can be attributed to the substantially improved film morphology and crystallinity (Supplementary Fig. 3) due to the retarded the crystallization of the perovskite (Supplementary Fig. 4). Fourier Transform infrared spectroscopy (FTIR) measurements revealed interaction between the BZM and the PbI_2 , which made the FAI have to compete with BZM to interact with the Pb-I , thus resulting in the sluggish crystallization kinetics of perovskites (Supplementary Fig. 5). Moreover, the introduction of BZM led to the formation of low-dimensional perovskite phases (Supplementary Fig. 6). These low-dimensional perovskite phases could effectively passivate defects in the perovskite film when residing at the grain boundaries as evidenced by the enhanced PL intensity (Supplementary Fig. 7) which suggested reduced non-radiative recombination of charge carriers.

Supplementary Fig. 4 | Photos of the perovskite films with and without BZM during the annealing process.

Supplementary Fig. 5 | FTIR spectra of pure BZM and BZM mixed with PbI_2 . The C=N stretching vibration, and the N-H stretching vibration of BZM shifted from 1672 to 1663 cm^{-1} , and from 3060 to 3048 cm^{-1} respectively, after mixing with PbI_2 , suggesting the binding of BZM to the Pb^{2+} through the amidinium group.

Supplementary Fig. 6 | Enlarged XRD patterns at low diffraction angles of the 1.4 M perovskite film with BZM.

Supplementary Fig. 7 | PL spectra of the 1.4 M perovskite thin films with and without introducing BZM.

Supplementary Fig. 48 | TRPL plots of 1.4 M perovskite with and without introducing BZM.”

2) Numerous organic salts have been employed as additives in perovskite films, with similar BZM compounds already having been used in perovskite solar cells. What role does BZM play in the perovskite film? Furthermore, does the concentration of BZM affect the performance of devices?"

Reply: We thank the reviewer’s detailed comments. As discussed above, we have performed additional experiments to provide further insights into the role of BZM in affecting the perovskite films. We performed FTIR measurements to investigate the interaction of BZM with perovskite precursors. As shown in **Fig. R6**, the C=N stretching vibration, and the N-H stretching vibration of BZM shifted from 1672 to 1663 cm^{-1} , and from 3060 to 3048 cm^{-1} respectively, after mixing with PbI_2 , suggesting the binding of BZM to the Pb^{2+} through the amidinium group. This interaction could slow down the crystallization of the perovskite because the FAI had to compete with BZM to interact with the Pb-I. The slower crystallization kinetics was indicated by the color change of the perovskite during the film fabrication process. The control film swiftly turned dark upon annealing (after ~ 2 seconds), whereas the film with BZM still

appeared a bit yellow till 4 seconds (**Fig. R7**). The sluggish crystallization process led to enlarged grain sizes (see the SEM images) and enhanced crystallinity as revealed by the higher diffraction intensity in the XRD measurements. From the XRD patterns of the BZM-based perovskite films, we also found that the introduction of BZM led to the formation of low-dimensional perovskite phases. This was backed by the XRD peaks at low diffraction angles (**Fig. R8**). These low-dimensional perovskite phases could effectively passivate defects in the perovskite film when residing at the grain boundaries as evidenced by the enhanced PL intensity which suggested reduced non-radiative recombination of charge carriers (**Fig. R9**).

Moreover, we managed to gain atomic-level insights into the structure of BZM-based low dimensional perovskite phase via single crystal analysis. Via an inverse temperature crystallization method, we successfully obtained the $(\text{BZM})_2\text{Pb}_3\text{I}_8$ single crystal that is the most thermodynamically favorable phase at room temperature. Different from most of the organic salts reported previously favoring the formation of two-dimensional perovskite phases, BZM delivered a one-dimensional perovskite phase with the Pb-I framework being isolated by the organic cations in one dimension (**Fig. R10**). Such one-dimensional structures afforded by the soft organic interaction could make their contacts with bulk 3D perovskites more flexible, thus more effectively mitigating the defects within the film. Moreover, the prior use of many bulky organic salts can lead to a large interlayer spacing of Pb-I, which can impede the carrier transport when they remain in the bulk. When the size of the organic tail goes larger, these additives can even be repelled towards the surfaces of the film. In such cases, these additives can only play a role in affecting the crystallization of perovskite films but cannot remain in the bulk film to further passivate the bulk defects. BZM in this study featured a small and conjugated tail unit, which can help minimize such concerns and benefit the carrier transport. We fully agree with the reviewer the concentration of BZM may significantly influence the carrier transport and thus the device performance. We have varied the concentration of BZM incorporated in the perovskite precursor and the resulted device performance was summarized in **Fig. R11**. It shows that the concentration of 4 mg/mL, led to the highest averaging PCE. Therefore, we used the optimized concentration of BZM in our study.

Owing to all the above-mentioned results, the BZM-based perovskite films we obtained exhibited long carrier lifetimes irrespective of film thickness, which enabled us to unveil the mechanisms restricting the performance of thick-film perovskite cells.

Fig. R6 FTIR spectra of pure BZM and BZM mixed with PbI_2 .

Fig. R7 Photos of the perovskite films with and without BZM during the annealing process.

Fig. R8 Enlarged XRD patterns at low diffraction angles of the 1.4 M perovskite film with BZM.

Fig. R9 PL spectra of the 1.4 M perovskite thin films with and without introducing BZM.

Fig. R10 Crystal structure of the 1D $(\text{BZM})_2\text{Pb}_3\text{I}_8$ perovskite.

Fig. R11 PCE statistics with different BZM concentrations in the 1.4 M perovskite thin films.

We have supplemented the data in Supplementary Materials and revised our manuscript accordingly as follows:

“We first successfully established a perovskite system with an extraordinarily long

carrier lifetime by introducing benzamidine hydrochloride (BZM) with an optimized concentration (Supplementary Fig. 1) into the perovskite precursor (**Fig. 1a**). Time resolved photoluminescence (TRPL) measurements revealed a typical carrier lifetime of 15.7 μ s.

Supplementary Fig. 1 | PCE statistics by varying BZM concentrations in 1.4 M.

The long carrier lifetime can be attributed to the substantially improved film morphology and crystallinity (Supplementary Fig. 3) due to the retarded the crystallization of the perovskite (Supplementary Fig. 4). Fourier Transform infrared spectroscopy (FTIR) measurements revealed interaction between the BZM and the PbI_2 , which made the FAI have to compete with BZM to interact with the Pb-I , thus resulting in the sluggish crystallization kinetics of perovskites (Supplementary Fig. 5). Moreover, the introduction of BZM led to the formation of low-dimensional perovskite phases (Supplementary Fig. 6). These low-dimensional perovskite phases could effectively passivate defects in the perovskite film when residing at the grain boundaries as evidenced by the enhanced PL intensity (Supplementary Fig. 7) which suggested reduced non-radiative recombination of charge carriers.

Supplementary Fig. 4 | Photos of the perovskite films with and without BZM during the annealing process.

Supplementary Fig. 5 | FTIR spectra of pure BZM and BZM mixed with PbI₂. The C=N stretching vibration, and the N-H stretching vibration of BZM shifted from 1672 to 1663 cm⁻¹, and from 3060 to 3048 cm⁻¹ respectively, after mixing with PbI₂, suggesting the binding of BZM to the Pb²⁺ through the amidinium group.

Supplementary Fig. 6 | Enlarged XRD patterns at low diffraction angles of the 1.4 M perovskite film with BZM.

Supplementary Fig. 7 | PL spectra of the 1.4 M perovskite thin films with and without introducing BZM.

Supplementary Fig. 49 | Crystal structure of the 1D (BZM)₂Pb₃I₈ perovskite.”

3) One key focus of this study revolves around the utilization of the SRS method for producing high-quality perovskite films. However, this part lacks in-depth discussions concerning the underlying reasons for this choice. Therefore, it becomes pertinent to delve into the factors that set the SRS method apart and elucidate its differentiating aspects.

Reply: We appreciate the reviewer’s valuable comments. We apologize for the ambiguity in our previous description. It is essential to clarify that the primary function of the SRS was to control and maintain strain within thick perovskite films, rather than produce high-quality perovskite films. Through the implementation of SRS, we successfully preserved the compressive strain within the thick perovskite films, which distinguished our SRS from other methods.

In our study, we found that the relaxed internal strain within a thick perovskite film adversely affected its conductivity. During the fabrication of perovskite thin films, the substrate can induce a temperature gradient during the annealing process, which contributed to the residual compressive strain within the lattice²⁻⁴. The gradual strain relaxation from the bottom to the top region across the film was verified by the GIWAXS measurements (**Fig. R12**). However, for the thick-film perovskite, the strain relaxed because the compressive strain was imposed by the bottom interface, the effect of which cannot reach that further when the film thickness increased. It is shown in **Fig. R13** that as the film thickness increased, the (001) diffraction peak of perovskite gradually shifted towards lower diffraction angles, verifying the gradual strain relaxation.

Hence, the design of SRS intended to maintain the strain in the thick perovskite film by creating additional interfaces that can induce strain step-by-step. The SRS involved a sequential deposition of perovskite precursors. After the first round of deposition, a thin perovskite layer formed through the reaction between the pre-deposited PbI₂ and the precursor solution containing organic cations. This thin

perovskite layer exhibited compressive strain within the lattice as the case of 1.4 M (please see the detailed discussions in our reply to the reviewer’s comment below). Therefore, it may serve as an intermediate layer for the subsequent deposition of the perovskite layer (**Fig. R14**). As the compressive strain was present in this intermediate layer, it could induce the lattice strain in the perovskite layer formed atop. The intermediate perovskite layer can be seen as a “quasi substrate” to induce lattice strain. As a result, compressive strain maintained in the thick perovskite film (SRS-2.0 M) as verified by the XRD measurements (**Fig. R15**), which finally led to a greatly improved PCE of 23.5% (certified 23.1%) for thick-film perovskite solar cells.

Fig. R12 Grazing-incident wide angle X-ray scattering profiles of 1.4 M perovskite films with the angle of incident beam ranging from 0.3° to 1.0°.

Fig.R13 X-ray diffraction patterns of perovskite films with different thicknesses.

Fig.R14 Schematic illustration of the mechanism of the SRS.

Fig. R15 XRD patterns of the (001) plane for the 1.4 M perovskite thin film, SRS-based 2.0 M perovskite thick film, and the intermediate perovskite layer (the perovskite film fabricated after the first round of dripping of the organic cation precursor in the SRS).

We have modified the manuscript and supplemented the data accordingly as follows:

“In light of the interface-induced compressive strain, the SRS we devised involved a layer-by-layer deposition of perovskite films which introduced additional interfaces for strain-inducing. The PbI_2 solution was first deposited onto the substrate, followed by the sequential deposition of the precursor solutions containing the organic cations. The first round of the precursor dripping led to the formation of a thin perovskite intermediate layer through the reaction between the pre-deposited PbI_2 and the precursor solution containing organic cations. (Supplementary Fig. 34-35). XRD measurements revealed that this thin intermediate perovskite layer exhibited the same compressive lattice strain as the case of 1.4 M (Supplementary Fig. 36). This layer can thus serve as a foundation for maintaining strain continuity during the subsequent second-round deposition of the perovskite precursor. This approach ensured that the compressive strain was maintained within each thin perovskite layer, which can be seen as a “quasi substrate” to induce lattice strain, and finally led to the formation of a thick perovskite layer with compressive strain in a manner akin to epitaxial growth.

Supplementary Fig. 35 | Schematic illustration of the mechanisms of SRS.

Supplementary Fig. 36 | X-ray diffraction patterns of the (001) plane for the 1.4 M perovskite thin film, SRS-based 2.0 M perovskite thick film, and the intermediate perovskite layer (the perovskite film fabricated after the first round of dripping of the organic cation precursor in the SRS).”

4) It's interesting that the authors observed the retention of compressive strain in the perovskite films fabricated using SRS. Why does this method make such a substantial difference in thick film devices? It is recommended that the authors provide further discussion, considering that readers would be highly interested in understanding the underlying reasons. Additionally, it is suggested to include supporting data for SRS-based films, such as SEM and XRD."

Reply: We thank the reviewer’s valuable comments. We have provided more results and discussions on the underlying mechanisms of the SRS for fabricating thick perovskite films. In our work, the compressive strain was generated due to the interface between perovskite and the substrate, as evidenced by the gradual strain relaxation from the bottom to the top region across the film (**Fig. R16**). The substrate induced temperature gradient during the annealing process, which contributed to the residual strain within the lattice²⁻⁴. When the film thickness becomes larger, the lattice strain starts to relax as the effects imposed by the interface gradually reduced.

In light of the origin of the compressive strain, we devised the SRS involving a layer-by-layer deposition of perovskite films which introduced additional interfaces for strain-inducing. The PbI_2 solution was first deposited onto the substrate, followed by the sequential deposition of the precursor solutions containing the organic cations. The first round of the precursor dripping led to the formation of a thin perovskite intermediate layer through the reaction between the pre-deposited PbI_2 and the precursor solution containing organic cations as depicted in **Fig. R17**. XRD measurements revealed that this thin intermediate perovskite layer exhibited compressive lattice strain as the case of 1.4 M (**Fig. R18**). This layer can thus serve as a foundation for maintaining strain continuity during the subsequent second-round deposition of the perovskite precursor. This approach ensured that the compressive strain was maintained within each thin intermediate perovskite layer, and finally led to the formation of a thick perovskite layer (SRS-2.0 M, **Fig. R18**) with compressive strain in a manner akin to epitaxial growth or a layer-by-layer deposition process.

As suggested by the reviewer, we also performed the SEM measurements on these films. Similar surface morphologies grain sizes were observed in the perovskite thin film (1.4 M), thick film (2.0 M), and thick film fabricated by SRS (SRS -2.0 M) as represented in **Fig. R19**. These results suggested that SRS would not change the surface morphology and the grain sizes of the perovskite film, and underscored the crucial influence of lattice strain in thick-film perovskite solar cells.

Fig. R16 Grazing-incident wide angle X-ray scattering profiles of 1.4 M perovskite films with the angle of incident beam ranging from 0.3° to 1.0° .

Fig.R17 Schematic illustration of how the strain can be maintained in the SRS-based perovskite thick films.

Fig. R18 XRD patterns of the (001) plane for the 1.4 M perovskite thin film, SRS-based 2.0 M perovskite thick film, and the intermediate perovskite layer (the perovskite film fabricated after the first round of dripping of the organic cation precursor in the SRS).

Fig. R19 Top-view SEM images of 1.4 M, 2.0 M and SRS-2.0 M based perovskite films.

We have modified the manuscript and Supplementary Materials accordingly as follows:

“In light of the interface-induced compressive strain, the SRS we devised involved a layer-by-layer deposition of perovskite films which introduced additional interfaces for strain-inducing. The PbI_2 solution was first deposited onto the substrate, followed by the sequential deposition of the precursor solutions containing the organic cations. The first round of the precursor dripping led to the formation of a thin perovskite

intermediate layer through the reaction between the pre-deposited PbI_2 and the precursor solution containing organic cations. (Supplementary Fig. 34-35). XRD measurements revealed that this thin intermediate perovskite layer exhibited the same compressive lattice strain as the case of 1.4 M (Supplementary Fig. 36). This layer can thus serve as a foundation for maintaining strain continuity during the subsequent second-round deposition of the perovskite precursor. This approach ensured that the compressive strain was maintained within each thin perovskite layer, which can be seen as a “quasi substrate” to induce lattice strain, and finally led to the formation of a thick perovskite layer with compressive strain in a manner akin to epitaxial growth.

Supplementary Fig. 35 | Schematic illustration of the mechanisms of SRS.

Supplementary Fig. 36 | X-ray diffraction patterns of the (001) plane for the 1.4 M perovskite thin film, SRS-based 2.0 M perovskite thick film, and the intermediate perovskite layer (the perovskite film fabricated after the first round of dripping of the organic cation precursor in the SRS).

Supplementary Fig. 50 | Top-view SEM images of 1.4 M, 2.0 M and SRS-2.0 M based perovskite films.”

5) The authors use two-step method to fabricate the perovskite films. So, the following description is not clear in the manuscript, “perovskite films were made by using perovskite precursor concentrations of 1.4, 1.5, 1.6, 1.8, and 2.0 M,”

Reply: We apologized for the unclear description and we have modified the sentence in the manuscript. In our study, perovskite films were fabricated through a two-step method. The perovskite films with different thickness were made by varying the concentration of PbI_2 to be 1.4, 1.5, 1.6, 1.8 and 2.0 M and the concentration of FAI/MACl proportionally adjusted accordingly.

We have revised our manuscript accordingly as follows:

“To systematically investigate the effects of film thickness on the optoelectronic properties of the films, we varied the thickness of the perovskite films by varying the concentration of PbI_2 to be 1.4, 1.5, 1.6, 1.8 and 2.0 M and the concentration of FAI/MACl proportionally adjusted accordingly. For convenience, the precursor concentrations, i.e. 1.4 M, 1.5 M, 1.6 M, 1.8 M, and 2.0 M, are used to represent the film thicknesses in this paper.”.

Reviewer #2 (Remarks to the Author):

In this work, the authors have chosen a very interesting topic, which is “strain regulates the photovoltaic performance of thick-film perovskites”. They ruled out the restrictions of carrier lifetime on the device performance by using additive BZM and delved into the critical role of the lattice strain in micron-scale thick perovskite films. Finally, an effective method to regulate strain has been applied to the fabrication of thick-film perovskite. In general, this paper has substantial data and impressive results. Thus, reviewer believe this work deserves being published in Nature Communications after revise.

Some comments are listed below for the further improvement:

Reply: We appreciate the reviewer’s comments and high praise of our work. The reviewer’s comments were replied point-by-point as follows.

1. In figure 2b, the authors present XRD patterns to study the difference of lattice structures. The (001) diffraction peak gradually shifted from $\sim 14.07^\circ$ to $\sim 14.01^\circ$ and this range is very small. And the authors have mentioned that “The peak shift to lower diffraction angles indicated an expansion of the perovskite lattice with increasing thickness, which could be attributed to a strain effect induced by the substrate.” What is the substrate of the XRD test samples? Is the XRD pattern of the substrate used for calibration during data processing? The authors described the XRD characterization too simply.

Reply: We thank the reviewer’s detailed comments, and we apologize for not including enough experimental details of the measurements. We have added more details of the XRD measurements and the corresponding data analysis. The resolution of our XRD measurement is 0.0001° , so the (001) diffraction peak gradually shifted from $\sim 14.07^\circ$ to $\sim 14.01^\circ$ can reliably demonstrate the expansion of the perovskite lattice. For the XRD measurements, the substrate of the samples was ITO glass with no characteristic XRD peaks. Prior to XRD analysis, all samples were positioned at the same height, aligned with the edge of the holder's top surface. The samples were subjected to a spin rate of 20 r/min to ensure averaged results. Consequently, any peak shifts can be attributed exclusively to strain within the films^{4,5}. In addition, at an incident beam angle of 1.0° , the GIWAXS data in **Fig. 2** also showed similar trend of peak shifts, which further verifies the reliability of the XRD results.

We have modified our manuscript as follows to include more details:

“The resolution of our XRD measurement is 0.0001° . Prior to XRD analysis, all samples were positioned at the same height, aligned with the edge of the holder's top surface. The samples were subjected to a spin rate of 20 r/min to ensure averaged results.”

2. This manuscript put forward strain regulation strategy (SRS), which achieved by sequential deposition of FAI/MACl solution on PbI₂ films. In this deposition method, whether the growth of perovskite crystals is controllable? Is the film preparation reproducible? Why can this deposition method implement the SRS strategy?

Reply: We thank the reviewer's professional comments. We have conducted characterizations including SEM, XRD and evaluated the PCEs of the as-fabricated devices for multiple times to cross check our results. These analyses demonstrated the high controllability of growth of perovskite crystals and the reproducibility of our film preparation process. Furthermore, we conducted additional experiments and detailed analysis to further elucidate the rationale of SRS in maintaining lattice strain, which also explain the controllability and reproducibility of this approach.

For perovskite films fabricated via SRS, their top-view SEM images all showed similar surface morphologies and grain sizes for different batches of samples (**Fig. R20**, suggesting high reproducibility of this approach. In the XRD measurements, all the samples from different batches showed similar diffraction patterns (**Fig. R21**). In addition, the devices with SRS also exhibited a consistent performance as evidenced by the statistical PCE results in **Fig. R22**. All these results verified that the SRS is controllable and reproducible.

The compressive strain was generated due to the interface between perovskite and the substrate, as evidenced by the gradual strain relaxation from the bottom to the top region across the film (**Fig. R23**). The substrate induced temperature gradient during the annealing process, which contributed to the residual strain within the lattice²⁻⁴. When the film thickness becomes larger, the lattice strain starts to relax as the effects imposed by the interface gradually reduced.

In light of the origin of the compressive strain, the SRS we designed involved a layer-by-layer deposition of perovskite films which introduced additional interfaces for strain-inducing. The PbI₂ solution was first deposited onto the substrate, followed by the sequential deposition of the precursor solutions containing the organic cations. The first round of the precursor dripping led to the formation of a thin perovskite intermediate layer through the reaction between the pre-deposited PbI₂ and the precursor solution containing organic cations as depicted in **Fig. R24**. XRD measurements revealed that this thin intermediate perovskite layer exhibited compressive lattice strain as the case of 1.4 M (**Fig. R25**). This layer can thus serve as a foundation for maintaining strain continuity during the subsequent second-round deposition of the perovskite precursor. This approach ensured that the compressive strain was maintained within each thin intermediate perovskite layer, and finally led to the formation of a thick perovskite layer (SRS-2.0 M, **Fig. 25**) with compressive strain in a manner akin to epitaxial growth or a layer-by-layer deposition process.

Fig. R20 Top-view scanning electron microscopy images of the SRS-2.0 M perovskite films from different batches.

Fig. R21 X-ray diffraction patterns of the SRS-2.0 M perovskite thin films from different batches.

Fig. R22 Statistics of efficiency parameters for SRS-2.0 M.

Fig. R23 Grazing-incident wide angle X-ray scattering profiles of 1.4 M perovskite films with the angle of incident beam ranging from 0.3° to 1.0°.

Fig.R24 Schematic illustration of the strain mechanisms of (A) SRS-2.0 M and (B) 1.4 M.

Fig. R25 X-ray diffraction patterns of the (001) plane for the 1.4 M perovskite thin film, SRS-based 2.0 M perovskite thick film, and the intermediate perovskite layer (the perovskite film fabricated after the first round of dripping of the organic cation

precursor in the SRS)

We have modified the manuscript and Supplementary Materials accordingly as follows:

“In light of the interface-induced compressive strain, the SRS we devised involved a layer-by-layer deposition of perovskite films which introduced additional interfaces for strain-inducing. The PbI_2 solution was first deposited onto the substrate, followed by the sequential deposition of the precursor solutions containing the organic cations. The first round of the precursor dripping led to the formation of a thin perovskite intermediate layer through the reaction between the pre-deposited PbI_2 and the precursor solution containing organic cations. (Supplementary Fig. 34-35). XRD measurements revealed that this thin intermediate perovskite layer exhibited the same compressive lattice strain as the case of 1.4 M (Supplementary Fig. 36). This layer can thus serve as a foundation for maintaining strain continuity during the subsequent second-round deposition of the perovskite precursor. This approach ensured that the compressive strain was maintained within each thin perovskite layer, which can be seen as a “quasi substrate” to induce lattice strain, and finally led to the formation of a thick perovskite layer with compressive strain in a manner akin to epitaxial growth.

Supplementary Fig. 35 | Schematic illustration of the mechanisms of SRS.

Supplementary Fig. 36 | X-ray diffraction patterns of the (001) plane for the 1.4 M perovskite thin film, SRS-based 2.0 M perovskite thick film, and the intermediate

perovskite layer (the perovskite film fabricated after the first round of dripping of the organic cation precursor in the SRS).

Supplementary Fig. 51 | Top-view scanning electron microscopy images of the SRS-2.0 M perovskite films from different batches.

Supplementary Fig. 52 | X-ray diffraction patterns of the SRS-2.0 M perovskite thin films from different batches.”

3. In figure 1f and Supplementary Fig.16, the performance of perovskite solar cells dropped as the increase of perovskite precursor concentration. Meanwhile, the authors described that “Perovskite films with various thicknesses of 0.80, 0.81, 0.87, 1.22, and 1.36 μm were made by using perovskite precursor concentrations of 1.4, 1.5, 1.6, 1.8, and 2.0 M, respectively (Supplementary Fig. 3).” As usually, as the film thickness increases, the absorbance of the film and the J_{sc} increase. This is contrary to Supplementary Fig.16b. Did the author consider this factor?

Reply: We thank the reviewer’s detailed comments. We fully agree with the reviewer that as the film thickness increases, it is expected that the absorbance of the film would also increase. However, the increased absorbance of the film does not necessarily lead to an increase in J_{sc} because J_{sc} is also affected by whether the photogenerated carriers can be efficiently transported towards the electrodes and finally get collected. In our study, we observed that as the thickness of the perovskite film increased, the conductivity of the film reduced (**Fig. R26**), which led to the reduction in the J_{sc} shown in Supplementary Fig.16b.

Fig. R26 The statistics of the conductivity of perovskite films collected by the conductive atomic force microscope.

4. Wrong description: “(FAI/MACl 80 mg/13 mg/4 mg in 1 mL IPA)”.

Reply: We Apologized for the typo and we have modified the manuscript accordingly as follows “For the perovskite layer fabricated by SRS, the PbI₂ solution was spin-coated on the substrate at 1500 rpm for 40 s, and then, the FAI/MACl solution (FAI/MACl 80 mg/13 mg in 1.7 mL IPA) was spin-coated on the PbI₂ film at 1800 rpm with sequential dripping of FAI/MACl/BZM (FAI/MACl/BZM 80 mg/13 mg/4 mg in 1 mL IPA) solution at 1800 rpm for 180 s.”

Reviewer #3 (Remarks to the Author):

This work investigates the underlying mechanisms for the low efficiency of thick-film based perovskite solar cells. By incorporating Benzamidine hydrochloride, the device performance is enhanced, which is attributed to the enhancement of mobility introduced compressive strain. The device performance is excellent, but the mechanisms are not surprising, which is revealed elsewhere, such as Nature 577, 209, Nature communications 10, 815, and <https://doi.org/10.1093/nsr/nwab047>.

Furthermore, when using XRD measurement to evaluate the strain states, it hypothesizes the composition of the film is uniform. However, based on the formula of this mixed perovskite, it is not sure if the compositional profiles over different samples with different thicknesses are the same, or their distributions are even. It is suggested to examine the final compositional information of different films, which is important to verify the conclusion of this work. Some additional comments are listed below for further improvement of this work.

Reply: We thank the reviewer's comments. We apologize that probably our original version of the manuscript was not clear enough and may cause some confusion. First of all, we would like to clarify that the enhancement of the performance of thick-film perovskite devices was not attributed to the enhancement of carrier mobility due to the incorporation of benzamidine hydrochloride (BZM). Instead, the incorporation of BZM was to achieve a sufficiently long carrier lifetime and diffusion length that enabled us to rule out the restrictions of carrier lifetime on the device performance. It is the incorporation of BZM that helped us elucidate the other underlying mechanism that restricted the device performance, but the BZM itself did not help with the PCE improvement in thick-film perovskite cells. We have modified our manuscript in this regard to make it clearer to the readers. We have also supplemented additional characterizations regarding compositional information to further consolidate the conclusion of this work.

We fully agree with the reviewer that many papers have studied the influence of strain on perovskite materials (such as the research papers and the reviewer pointed out by the reviewer). Lattice strain is indeed well-known as an important factor in affecting the properties of perovskites and actually almost all kinds of materials. However, we think that understanding strain as an important factor that affects material properties does not necessarily mean that the strain-involved mechanism we proposed is not surprising. It is how the strain affect what kind of material property that makes a difference and contribute to the conceptual advances of a study. Mechanisms that restrict the performance of thick-film perovskite have yet to be elucidated before, which is the focus of our study. In Nature 577, 209, it demonstrated the strained epitaxial growth of halide perovskite single-crystal on lattice-mismatched halide perovskite substrates, with emphasizing on the crystal structure, electronic properties and stabilization effect on the α -FAPbI₃ phase. Nevertheless, our study uniquely centered on elucidating the previously overlooked mechanisms responsible for the decline in

PCE in thick perovskite solar cells, addressing a critical gap in the current research landscape. In Nature Communications 10, 815, the authors identified a gradient distribution of in-plane strain component that correlated to the composition variations perpendicular to the substrate for the perovskite thin films with a thickness of $\sim 0.5 \mu\text{m}$, and how it impacted the carrier dynamics. However, none of these studies have touched on the polycrystalline perovskite thick films and the restricting factors of their solar cell devices, and let alone how to solve this problem. Therefore, we think those papers about the roles of strain in affecting the properties of perovskite thin films and single crystals are not relevant to the results and conclusions of our study. Moreover, beyond elucidating of the mechanism restricting the performance of thick-film perovskite solar cells, we developed a strain regulation strategy to address this issue and achieve highly efficient thick-film perovskite solar cells.

Per the reviewer's other comments, we have provided a point-to-point response below with additional data and analysis.

1. Page 6 line 173: In Figure 2b, the description of “the peak shift to lower diffraction angles indicated an expansion of the perovskite lattice” may not be accurate. The peak shift to lower diffraction angles indicated an increased spacing of the crystal plane parallel to the substrate, how about the lattice spacing perpendicular to the substrate?

Reply: We thank the reviewer's valuable comments. We highly agree with the reviewer that, in a perovskite single crystal, the peak shift to lower diffraction angles indeed indicated an increased spacing of a specific crystal plane parallel to the substrate. However, our perovskite films are polycrystalline. Therefore, XRD has been widely employed to illustrate the lattice strain within the polycrystalline perovskite films, for example:

“...compared with $(\text{FAPbI}_3)_{0.972}(\text{MDACl}_2)_{0.038}$ (denoted as control), the peak position at $\sim 14^\circ$ gradually shifted to higher angles from 14.07° to 14.16° as x increased to 0.03, then it slightly decreased to 14.12° at $x=0.04$ (Fig. 1C). In the same crystal, because the diffraction angle reflected the expansion and contraction of the lattice...” in Kim et al., Science 370, 108–112 (2020);

“To rationalize the above slow-passivation phenomena observed for the $\text{FASnI}_3\text{-EDAI}_2$ 1% devices, the corresponding films were studied with XRD and PL decays; the corresponding results are shown in Fig. 4d and e, respectively. For XRD, the diffraction signals related to the (h00) planes showed almost no shift but those for directions such as facets (120) and (222) shifted to the small-angle region (Fig. 4d). The shifts in XRD as a function of storage duration might be related to the relaxation of strain due to the crystal stress originally existing in the fresh sample, leading to an increased d-spacing values and volume of the unit cell” in Energy Environ. Sci. 11.9 (2018): 2353-2362;

“We hypothesize that phase transformation from the α - to δ -phase in FAPbI_3 is the guiding factor. On the basis of our results, we found that MABr alloying has two effects on structure: reducing the lattice volume and relaxing the strain in lattice. From the

XRD pattern shown in Figure 2a, peaks shift toward higher angles after MABr alloying, indicating shrinkage of the lattice according to the Braggs law, $n\lambda=2d_{hkl}\sin\theta$ in ACS Energy Lett. 2016, 1, 1014–1020.

Nevertheless, to reflect the reviewer’s comment, we integrated the in-plane diffraction patterns from the GIWAXS data with the azimuth angle ranging from 0° to 10° . (Fig. R27). The GIWAXS peaks along the in-plane direction exhibited negligible shift (0.975 \AA^{-1} for 1.4 M and 0.976 \AA^{-1} for 2.0 M perovskite films), suggesting little in-plane lattice strain. Therefore, the lattice expansion mainly occurred in the out-of-plane direction of the perovskite films.

Fig. R27 Integrated GIWAXS patterns with the azimuth angle ranging from 0° to 10° for the 1.4 M and 2.0 M perovskite films.

We have modified the manuscript and Supplementary Materials accordingly as follows:

Supplementary Fig. 53 | Integrated GIWAXS patterns with the azimuth angle ranging from 0° to 10° for the 1.4 M and 2.0 M perovskite films. The GIWAXS peaks along the in-plane direction exhibited negligible shift (0.975 \AA^{-1} for 1.4 M and 0.976 \AA^{-1} for 2.0 M perovskite films).”

2. Perovskite films' strain state (tension or compressive) should be checked by referencing the strain-free state (freestanding perovskite powder scraped from the as-prepared perovskite films from the substrates) or with “ $[\sin]^2 \psi$ method”. The

GIWAXS patterns are suggested to be added to the main text or Supplementary Materials, which could be in favor of the analysis of the strain of perovskite film according to the Qz value of different azimuths.

Reply: We thank the reviewer's valuable comments. We have performed additional XRD measurements for the perovskite powders to confirm the strain state in the perovskite thin films. We compared the XRD patterns of the perovskite powders that scraped from the substrates and the perovskite films (**Fig. R28**). For the free-standing perovskite powders scraped from the substrate, a diffraction peak of (001) plane was located at 13.98° , which showed a shift towards lower diffraction angle compared to that of the perovskite thin film (14.07°). Therefore, the strain state of the perovskite film was confirmed to be compressive.

Fig. R28 XRD patterns of the 1.4 M perovskite film and the 1.4 M perovskite powders scraped from the substrate.

As suggested by the reviewer, we have also appended the full GIWAXS patterns in the Supplementary Materials as follows:

“**Supplementary Fig. 57** | GIWAXS patterns of the 1.4 M and 2.0 M perovskite films (The number after the dash sign in the right upper corner denotes the incident angle).”

3. The examination of the film chemistry to rule out any influence from composition variation in thin and thick perovskite films should be more careful. Depth-resolved X-ray photoelectron spectroscopy (XPS) and/or other characterization are suggested to conform to the composition variation.

Reply: We thank the reviewer’s valuable comments. We apologize for the unavailability of depth-resolved X-ray photoelectron spectroscopy (XPS) measurements in our institutions recently. As an alternative, we employed time-of-flight secondary ion mass spectrometry (SIMS) to unveil the distribution of chemical compositions within the film (**Fig. R29**). The distribution profiles of Pb^{2+} , Cs^+ , FA^+ , and In^+ (as a reference to indicate the interface) were displayed in **Fig. R29**. The

perovskite thick film showed very similar composition distribution to that of its thin film counterpart. This helped rule out the effect of the compositional distribution on the device performance, and further verified our conclusion.

Fig. R29 Time of Flight secondary ion mass spectrometry of (A) 1.4 M and (B) 2.0 M perovskite films. ToF-SIMS depth-profile analysis was performed using a PHI NanoTOF III instrument (ULVAC-PHI, Inc.), where a 3 kV Ar ion beam was used for erosion and a 25 keV Bi⁺ pulsed primary ion beam was used for the analysis. The area of analysis was 100 × 100 μm² while the sputtering area was 400 × 400 μm².

We have modified the manuscript and supplemented the data accordingly as follows:

“We also performed time-of-flight secondary ion mass spectrometry (SIMS) to study the distribution of chemical compositions throughout the film. The distribution profiles of Pb²⁺, Cs⁺, FA⁺, and In⁺ (as a reference to indicate the interface) were displayed in Supplementary Fig. 24. The perovskite thick film showed very similar composition distribution to that of its thin film counterpart. These results ruled out the effect of the compositional distribution on the device performances.

Supplementary Fig. 24 | Time of Flight secondary ion mass spectrometry of (A) 1.4 M and (B) 2.0 M perovskite films. ToF-SIMS depth-profile analysis was performed using a PHI NanoTOF III instrument (ULVAC-PHI, Inc.), where a 3 kV Ar ion beam was used for erosion and a 25 keV Bi⁺ pulsed primary ion beam was used for the analysis. The area of analysis was 100 × 100 μm² while the sputtering area was 400 ×

400 μm^2 .”

4. It is interesting to see the significant effect of strain regulation on electron mobility and electron concentration in perovskite. Does this strategy have broader implications, such as how it works with different components?

Reply: We thank the reviewer’s valuable comments. As suggested by the reviewer, we applied the SRS to a different wide-bandgap (WB) perovskite composition ($\text{Cs}_{0.05}\text{FA}_{0.8}\text{MA}_{0.15}\text{PbI}_{2.25}\text{Br}_{0.75}$). **Fig. R30** shows the XRD patterns of the WB perovskite thick films fabricated with and without SRS. The main diffraction peak shifted to lower diffraction angles, from 14.31° in WB-2.0 M to 14.47° in WB-SRS-2.0 M, indicative of the compressive strain with the SRS, akin to that observed in the narrow-bandgap sample. Moreover, compared to the WB 2.0 M perovskite thick film fabricated without the SRS, the electron concentration in the WB-SRS-2.0 M film increased and the electron mobility decreased. (**Fig. R31-32**), which is highly consistent with the trends observed in the narrow-bandgap perovskite composition.

Beyond the reviewer’s suggestion, we have also tested the applicability of this strategy to different substrates. When deposited on the textured Si, the film fabricated with the SRS showed a XRD peak shift towards a higher diffraction angle (from 14.27° in 2.0 M to 14.31° in SRS-2.0 M), suggesting the compressive strain in SRS-2.0 M (**Fig. R33**). Therefore, our strategy indeed has broader implications in terms of different perovskite components and substrates.

Fig. R30 XRD patterns of 2.0 M and SRS-2.0 M for wide-bandgap perovskite films.

Fig. R31 Statistics of electron concentration of wide bandgap perovskite derived from

capacitance-voltage (Mott-Schottky) plot.

Fig. R32 Statistics of electron mobility of wide bandgap perovskite extracted from space-charge-limited-current (SCLC) measurement.

Fig. R33 (001) plane of XRD patterns of SRS-2.0 M and 2.0 M that deposited on the textured Si.

We have added the morphology analysis in the Supplementary Materials as follows:

Supplementary Fig. 58 XRD patterns of 2.0 M and SRS-2.0 M for wide-bandgap perovskite films.

Supplementary Fig. 59 Statistics of electron concentration of wide bandgap perovskite derived from capacitance-voltage (Mott-Schottky) plot.

Supplementary Fig. 60 Statistics of electron mobility of wide bandgap perovskite extracted from space-charge-limited-current (SCLC) measurement.

Supplementary Fig. 61 (001) plane of XRD patterns of SRS-2.0 M and 2.0 M that deposited on the textured Si.”

- 1 Ni, Z. *et al.* Resolving spatial and energetic distributions of trap states in metal
halide perovskite solar cells. *Science* **367**, 1352-1358, (2020).
- 2 Liu, D. *et al.* Strain analysis and engineering in halide perovskite photovoltaics.
Nat. Mater. **20**, 1337-1346, (2021).
- 3 Chen, Y. *et al.* Strain engineering and epitaxial stabilization of halide
perovskites. *Nature* **577**, 209-215, (2020).
- 4 Zhu, C. *et al.* Strain engineering in perovskite solar cells and its impacts on
carrier dynamics. *Nat. Commun.* **10**, 815, (2019).
- 5 Kim, G. *et al.* Impact of strain relaxation on performance of α -formamidinium
lead iodide perovskite solar cells. *Science* **370**, 108-112, (2020).
- 6 Miyano, K., Tripathi, N., Yanagida, M. & Shirai, Y. Lead Halide Perovskite
Photovoltaic as a Model p-i-n Diode. *Acc. Chem. Res.* **49**, 303-310, (2016).
- 7 Edri, E. *et al.* Elucidating the charge carrier separation and working mechanism
of CH₃NH₃PbI₃-xCl_x perovskite solar cells. *Nat. Commun.* **5**, 3461, (2014).
- 8 Uratani, H. & Yamashita, K. Charge carrier trapping at surface defects of
perovskite solar cell absorbers: a first-principles study. *The J. Phy. Chem. Lett.*
8, 742-746 (2017).
- 9 Yin, W.-J., Shi, T. & Yan, Y. Unusual defect physics in CH₃NH₃PbI₃ perovskite
solar cell absorber. *Appl. Phy. Lett.* **104** (2014).
- 10 Kim, H.-S. & Park, N.-G. Importance of tailoring lattice strain in halide
perovskite crystals. *NPG Asia Mater.* **12**, 78 (2020).

REVIEWER COMMENTS

Reviewer #1 (Remarks to the Author):

The authors have addressed the reviewer's comments carefully. I think it should be accepted for publication in Nature Communications.

Reviewer #2 (Remarks to the Author):

The manuscript reports an interesting research using the strain to regulate the properties of thick perovskite film and the corresponding photovoltaic performance of PSCs. The authors present an insightful mechanism for the performance reduction of thick-film PSCs that focuses on lattice strain rather than the carrier lifetime of perovskite films. They identify that the thicker perovskite film could be desirable in tandem devices, more stable application, and scalable fabrication. After a round of revision, the quality of the manuscripts has undoubtedly improved, and the authors have made an appropriate response. Now, there are still several questions that need to be discussed.

(1) It is a critical approach in this manuscript using the GIWAXS to analyze the strain information in perovskite film since GIWAXS can profile the information of perovskite film at different depths by varying the incident beam angle. Considering the thicknesses of films with 1.4 M ($\sim 0.8 \mu\text{m}$), 2.0 M ($\sim 1.4 \mu\text{m}$) and SRS-2.0M ($\sim 2.1 \mu\text{m}$) varies greatly, it is difficult to characterize the strain information at the whole perovskite film using the same angle. For example, using the angle of 1.0° could detect the whole film of 1.4 M, but maybe only half the thickness of the SRS-2.0M film.

(2) It is pretty interesting that the manuscripts revealed that the strain is the crucial factor determining the conductivity of perovskite film. The type and distribution of strain are closely related to the depth of perovskite film, which is verified by the GIWAXS results and previous reports. It is recommended to supplement schematic diagrams to exhibit the existing strain and its type and distribution in the perovskite film with different thicknesses, which can help the readers understand more clearly.

(3) In lines 162 and 163, the authors proposed that the increased series resistance (from 38.5Ω to 116.0Ω) is the reason for the reduced fill factor (from 0.77 to 0.45) and decreased PCE (from 23.1% to 10.9%). The increased series resistance results from the reduced conductivity of perovskite film. In lines 305 and 306, after using the SRS strategy, the authors successfully make the conductivity of SRS-2.0 M similar to that of 1.4 M, which achieves an average PCE of 22.9%, fill factor of 0.75 and series resistance of 38.5Ω . Actually, the resistance is determined not only by the conductivity but also by the thickness. Based on the similar conductivity, it is confusing that the authors obtained the same series resistance while the thickness increased from $0.8 \mu\text{m}$ (1.4 M) to $2.1 \mu\text{m}$ (SRS-2.0 M).

Reviewer #3 (Remarks to the Author):

We thank the authors for the efforts made in replying to the comments and revising the manuscript. The manuscript focused on enhancing the power conversion efficiency of perovskite solar cells with thick absorbers. While one of the motivations for using thick absorbers is to enhance the stability of devices. The authors are recommended to study the stability of perovskite solar cells as a function of perovskite thickness, referring to the ISOS protocols (Nat. Energy, 2020, 5, 35).

RESPONSE LETTER (NCOMMS-23-30941)

Reviewer #1 (Remarks to the Author):

The authors have addressed the reviewer's comments carefully. I think it should be accepted for publication in Nature Communications.

REPLY: We thank the reviewer's support on the publication of our work in Nature Communications.

Reviewer #2 (Remarks to the Author):

The manuscript reports an interesting research using the strain to regulate the properties of thick perovskite film and the corresponding photovoltaic performance of PSCs. The authors present an insightful mechanism for the performance reduction of thick-film PSCs that focuses on lattice strain rather than the carrier lifetime of perovskite films. They identify that the thicker perovskite film could be desirable in tandem devices, more stable application, and scalable fabrication. After a round of revision, the quality of the manuscripts has undoubtedly improved, and the authors have made an appropriate response. Now, there are still several questions that need to be discussed.

REPLY: We appreciate the reviewer's praise on the revised version of our manuscript, as well as the additional questions. We have provided a point-to-point response below to address the reviewer's questions.

(1) It is a critical approach in this manuscript using the GIWAXS to analyze the strain information in perovskite film since GIWAXS can profile the information of perovskite film at different depths by varying the incident beam angle. Considering the thicknesses of films with 1.4 M (~0.8 μm), 2.0 M (~1.4 μm) and SRS-2.0M (~2.1 μm) varies greatly, it is difficult to characterize the strain information at the whole perovskite film using the same angle. For example, using the angle of 1.0° could detect the whole film of 1.4 M, but maybe only half the thickness of the SRS-2.0M film.

Reply: We thank the reviewer's detailed comments. As a matter of fact, we used XRD measurements to evaluate the overall lattice strain in the as-fabricated perovskite thin films, which is the main evidence supporting our conclusions. XRD measurements are performed on sealed-tube Cu X-ray source, equipped with 1D LynxEye detector. The X-ray of XRD pattern could penetrate all the films in this work. As shown in **Fig. R1**, all samples exhibited two dominant diffraction peaks at around 14° and 28° , which can be assigned to the characteristic (001) and (002) planes of the α -phase perovskite. The (001) diffraction peaks downwards shifted from $\sim 14.07^\circ$ to $\sim 14.01^\circ$ as the film thickness increased from 1.4 M to 2.0 M. In contrast, the (001) diffraction peaks in the SRS-2.0 M upwards shifted to $\sim 14.10^\circ$. The peak shift to larger diffraction angle verified the lattice compression in the SRS-2.0 M film similar to the 1.4 M thin film.

The depth resolved GIWAXS, instead, was used to investigate the strain distribution

within the films, which only helped to support that the compressive lattice strain was induced by the bottom substrate. That said, for both the 1.4 M and 2.0 M perovskite samples, using the angle of 1.0° can detect the whole films as indicated by the detected signals of the substrates (**Fig. R2**). For the SRS-2.0 M, we fully agree with the reviewer that the beam at an angle of 1.0° could not penetrate the whole film, but at least $\sim 70\%$ of the film can be detected. This depth was enough for us to observe the compressive strain near the bottom side of the film. Therefore, in combination with the XRD results showing the strain information of the whole film, the retained compressive lattice strain via the SRS can be verified.

Fig. R1 X-ray diffraction patterns at (a) (001) planes and (b) (002) planes of 1.4 M, 2.0 M and SRS-2.0 M.

Fig. R2 GIWAXS diffraction pattern at an incident angle of 1.0° of 1.4 M and 2.0 M. The signals of substrate were marked by white circles in the figures.

(2) It is pretty interesting that the manuscripts revealed that the strain is the crucial factor determining the conductivity of perovskite film. The type and distribution of strain are closely related to the depth of perovskite film, which is verified by the GIWAXS results and previous reports. It is recommended to supplement schematic diagrams to exhibit the existing strain and its type and distribution in the perovskite film with different thicknesses, which can help the readers understand more clearly.

Reply: We thank the reviewer's valuable comments. We supplemented schematic diagrams (**Fig. R3**) to better illustrate the strain information within different kinds of

perovskite films, and the picture has been added in the supplementary materials.

Fig. R3 Schematic illustration of the strain type and distribution in the perovskite films of 1.4 M, 2.0 M and SRS-2.0 M.

(3) In lines 162 and 163, the authors proposed that the increased series resistance (from 38.5Ω to 116.0Ω) is the reason for the reduced fill factor (from 0.77 to 0.45) and decreased PCE (from 23.1% to 10.9%). The increased series resistance results from the reduced conductivity of perovskite film. In lines 305 and 306, after using the SRS strategy, the authors successfully make the conductivity of SRS-2.0 M similar to that of 1.4 M, which achieves an average PCE of 22.9%, fill factor of 0.75 and series resistance of 38.5Ω . Actually, the resistance is determined not only by the conductivity but also by the thickness. Based on the similar conductivity, it is confusing that the authors obtained the same series resistance while the thickness increased from $0.8 \mu\text{m}$ (1.4 M) to $2.1 \mu\text{m}$ (SRS-2.0 M).

Reply: We thank the reviewer's detailed comments. According to the statistical results of the conductivity measurements, the average conductivity of the film fabricated with the SRS-2.0 M was 0.0060 S/cm , which was actually higher than that in the 1.4 M perovskite thin film (0.0055 S/cm). This could compensate for the electrical loss resulting from the increased film thickness. More importantly, the conductivity we derived from the conductive AFM measurements was based on perovskite films instead of perovskite devices. Therefore, the absolute value of the conductivity cannot reflect the overall conductivity of a full device. In contrast, the series resistance extracted from the $J-V$ curves characterizes the property of the device as a whole, and is thus influenced by various factors, such as the transporting layers and interfacial contacts. As a result, the conductivity of the perovskite films we obtained is related to, but is not necessarily to be linearly related to the series resistance of a whole device. This rationalizes the similar series resistance we observed in the 1.4 M and SRS-2.0 M perovskite despite the difference in the film thickness.

Reviewer #3 (Remarks to the Author):

We thank the authors for the efforts made in replying to the comments and revising the manuscript. The manuscript focused on enhancing the power conversion efficiency of perovskite solar cells with thick absorbers. While one of the motivations for using thick absorbers is to enhance the stability of devices. The authors are recommended to study the stability of perovskite solar cells as a function of perovskite thickness, referring to the ISOS protocols (Nat. Energy, 2020, 5, 35).

Reply: We thank the reviewer for the valuable suggestions. We have indeed followed the ISOS-L-1 protocol and performed the stability tests on the perovskite devices with different perovskite thicknesses (included in the supplementary materials). As shown in **Fig. R4**, the device with a 2.05 μm perovskite layer maintained over 85% of its initial PCE after constant illumination at AM 1.5G for ~ 1500 h at open-circuit condition. However, the device with a 0.80 μm perovskite layer lost over 50% of its initial PCE with the same aging time. Hence, the application of a thick perovskite film proved advantageous in enhancing device stability.

Fig. R4 The PCE tracking, following the ISOS-L-1 protocol, of encapsulated perovskite devices with different perovskite thicknesses aged at open-circuit condition under continuous illumination (AM 1.5G) and 50% relative humidity.

REVIEWERS' COMMENTS

Reviewer #2 (Remarks to the Author):

I have carefully read the author's revision reply and the main text of the manuscript, and I find that the author has modified according to the suggestions. The authors further improved the manuscript. All my concerns have been addressed. So, in my mind, it can be accepted.

Reviewer #3 (Remarks to the Author):

This revision has addressed my previous concerns. I believe it is ready for publication.